



# The EUMETSAT sea ice climate record

R. T. Tonboe[1], S. Eastwood[2], T. Lavergne[2], A. M. Sørensen[2], N. Rathmann[3], G. Dybkjær[1], L. Toudal Pedersen[1], J. L. Høyer[1], S. Kern[4]

[1]Danish Meteorological Institute, Lyngbyvej 100, Copenhagen, DK-2100, Denmark
[2]Norwegian Meteorological Institute, P.O.BOX 43, Oslo, N-0313, Norway
[3]University of Copenhagen, Juliane Maries Vej 30, Copenhagen, DK-2100, Denmark
[4]University of Hamburg, Grindelberg 5, Hamburg, D-20144, Germany

*Correspondence to*: R. T. Tonboe (rtt@dmi.dk)

**Abstract.** An Arctic and Antarctic sea ice area and extent dataset has been generated by

EUMETSAT's Ocean and Sea Ice Satellite Application Facility (OSISAF) using the record of

American microwave radiometer data from Nimbus 7 Scanning Multichannel Microwave radiometer

(SMMR) and the Defense Meteorological satellite Program (DMSP) Special Sensor Microwave/

Imager (SSM/I) and Special Sensor Microwave Imager and Sounder (SSMIS) satellite sensors. The

dataset covers the period from 1978 to 2014 and updates and further developments are planned for the

next phase of the project. The methodology is using: 1) numerical weather prediction (NWP) input to a

radiative transfer model (RTM) for correction of the brightness temperatures for reduction of

atmospheric noise, 2) dynamical algorithm tie-points to mitigate trends in residual atmospheric, sea ice

and water emission characteristics and inter-sensor differences/biases, 3) and a hybrid sea ice

concentration algorithm using the Bristol algorithm over ice and the Bootstrap algorithm in frequency

mode over open water. A new algorithm has been developed to estimate the spatially and temporally

varying sea ice concentration uncertainties. A comparison to sea ice charts from the Arctic and the

Antarctic shows that ice concentrations are higher in the ice charts than estimated from the radiometer

data at intermediate ice concentrations. The sea ice climate dataset is available for download at

(www.osisaf.org) including documentation.

## 1. Introduction

The Arctic sea ice area and extent has decreased since the 1970s (Cavalieri and Parkinson, 2012). In

Antarctica there are large regional differences in trends but overall the sea ice extent is increasing

because of changing atmospheric circulation patterns and regional cooling (Comiso et al., 2011). The

climatic trends in sea ice extent have been documented using models (Zhang and Walsh, 2006), ice

charts (Rayner et al., 2003) and in particular the passive microwave data record from American satellite





microwave radiometers (Parkinson and Cavalieri, 2012; Cavalieri and Parkinson, 2012). Here the sea
ice extent is defined as ice covered waters with ice concentrations derived from microwave radiometer
data greater than 15% as in Parkinson and Cavalieri (2008) and at a grid resolution of 12.5 x 12.5
kilometers.
The brightness temperatures measured by the satellite radiometers at the atmospheric window channels
are dominated by surface emission. However, the measured brightness temperatures are also affected
by atmospheric parameters such as wind roughening of the ocean surface, water vapor and cloud liquid
water (Wentz, 1983 and 1997; Andersen et al., 2006B). These parameters have trends over the
observing period (Wentz et al., 2007). Even though the sensitivity to these parameters is minimized in
ice concentration algorithms in general different algorithms still have different sensitivities resulting in
structural uncertainties, i.e. different outcome from different algorithms using the same data (Andersen
et al., 2006B).
Because of the algorithms different sensitivities to the noise, and that the noise has climatic trends, the
differences are also reflected in the sea ice extent trends (Andersen et al., 2007). To minimize these
artificial trends we must: 1) find algorithms with low sensitivities to the atmospheric and surface
emissivity variability, 2) correct the brightness temperatures for the properties that we are able to
quantify, and in particular when doing this it is important to 3) calibrate the algorithms to the actual ice
and water signatures using dynamical tie-points, and finally 4) quantify the residual uncertainties. The
EUMETSAT sea ice climate record (ESICR) is generated according to these principles, 1 - 4, and it is
based on the Nimbus 7 Scanning Multichannel Microwave Radiometer (SMMR) (1978-1987), the
Special Sensor Microwave/Imager (SSM/I) (1987-2009) and the Special Sensor Microwave Imager and
Sounder (SSMIS) (2003-today) radiometer data. It uses a combination of the Bristol (Smith, 1996) and
the Bootstrap (Comiso, 1986) algorithms with dynamical tie-points, explicit atmospheric correction
using numerical weather prediction data for error reduction and it comes with spatially and temporally
varying uncertainty estimates describing the residual uncertainties.
Uncertainty estimates are needed when the ice concentration data are compared to other data sets or
when the ice concentrations are assimilated into numerical models. The mean accuracy of some of the



more common algorithms, used to compute ice concentration from SSM/I data, such as the NASA
Team and Bootstrap are reported to be 1-6% in winter (Steffen and Schweiger, 1991; Emery et al.,
1994; Belchansky and Douglas, 2002). The overall accuracy of the SMMR total ice concentrations is
estimated to be ±7% (Gloersen et al., 1992). During summer the uncertainties are larger than during
winter (Ivanova et al., 2015).
**1.1 Description of the Nimbus 7 SMMR instrument and data**
The SMMR instrument on board the Nimbus 7 satellite operated from October 1978 to August 1987
(Gloersen et al., 1992). The instrument had 10 channels from the six Dicke radiometers at five
frequencies (6.6, 10.7, 18.0, 21.0, 37.0 GHz) and vertical and horizontal linear polarization. The across
track scanning was accomplished by tilting the reflector from side to side while maintaining a constant
incidence angle on the ground of about 50.2°. The scan track on the ground formed a 780 km wide arc
in front of the satellite (Gloersen and Barath, 1977). Because of the satellite orbit inclination and swath
width there is no coverage pole-wards of 84°. There is SMMR data only every second day. Data were
provided by the National Snow and Ice Data Center (NSIDC) as brightness temperatures in swath
"projection" (Meier, 2008).
**1.2 Description of the SSM/I and SSMIS instruments and data.**
The SSM/I instruments onboard the Defense Meteorological Satellite Program (DMSP) are conically
scanning instruments with 7 total power radiometers measuring at 19.35v, 19.35h, 22.2h, 37.0v, 37.0h,
85.5v, and 85.5h. The incidence angle is 53.1 degrees and the swath width on the Earth's surface is
about 1400 km. There is no coverage pole-wards of 87 degrees. The different satellites and their
operation periods are listed in Table 2. The SSM/I data (version 6) was purchased by EUMETSAT
from Remote Sensing Systems (RSS) as antenna temperatures and converted to brightness
temperatures using RSS software. The Remote Sensing Systems (RSS) SSM/I version 6 post
processing includes geo-location correction, sensor calibration and quality control procedures, and inter
calibration between the different satellites from overlapping periods. These procedures are documented
in the RSS SSM/I User's Manuals (Wentz, 1991; Wentz, 1993; Wentz, 2006).
The SSMIS is a continuation of the SSM/I series of instruments onboard the DMSP satellites but with



an extension in the number of channels. SSMIS has 24 channels between 19 and 183 GHz. The 19 and
37 GHz channels which are used in the ESICR have identical frequencies on SSM/I and SSMIS.
However, SSMIS has a swath width of about 1700km which gives near complete daily coverage of the
Arctic Ocean. The SSMIS data are from the L2B near real time data-stream.
**1.3 Meteorological data**
The Numerical Weather Prediction (NWP) model meteorological data are used for reduction of the
brightness temperatures for atmospheric noise with a radiative transfer model. European Centre for
Medium-range Weather Forecast (ECMWF) ERA 40 data are used for the period from 1978 to 2002,
and ECMWF data from the operational models are used from 2002 onwards. A description of the ERA
40 meteorological data archive and the reanalysis can be found in Kålberg et al. (2004). We use 6
hourly data at a resolution of 1.25 degrees.
**1.4 MODIS data**
The coarse resolution of the brightness temperature measurements gives rise to an additional
uncertainty when sea ice concentration is reproduced at finer grid spacing. We call this smearing
uncertainty and it is estimated using a smearing model (see section 2.9). High resolution ice
concentration data are used as input to the smearing model. Cloud free MODIS scenes from the NASA
image gallery archive (http://rapidfire.sci.gsfc.nasa.gov/cgi-bin/imagery/gallery.cgi) were selected
manually for their different sea ice conditions: low concentration, medium and high concentration.
Parts of the image with cloud cover were cut out. The band 1 (620 - 670 nm) brightness is high -
typically greater than 220 for sea ice and less than 60 for open water. These two upper and lower
values are used for scaling pixels between 100% and 0% ice concentration respectively. Pixels with
intermediate brightness are assigned intermediate concentrations linearly. Brightness above 220 and
below 60 is truncated to 100% and 0% respectively. The 250 m spatial resolution is re-sampled to 1 km
pixel resolution.
**1.5 Ice chart data for comparison**
The operational sea ice charts from the National Ice Center (NIC) are a relatively independent source
of ice information (not necessarily unbiased) for comparing to the sea ice concentration estimates. The



ice charts, intended for aiding navigation are produced on a regular basis covering all seasons, both
Southern and Northern hemispheres and the time series cover the entire climate record period except
for the period Dec. 1994 to Jan. 2003 in the Southern hemisphere where we have been unable to
acquire digital ice charts.
Ice charts are produced manually on the basis of a multitude of satellite and reconnaissance data for
ship navigation support. The ice charts are detailed manual interpretations of primarily satellite
imagery and a subsequent mapping procedure is carried out by ice analysts. The ice charts are primarily
used for strategic and tactical planning within the offshore and shipping community.
The more recent ice charts are based partly on satellite SAR data e.g. RADARSAT 1 since 1995 and
ENVISAT since 2002, various scatterometers together with visual/infrared line scanners e.g. AVHRR,
MODIS, OLS whenever possible for daylight and cloud cover conditions. Also the passive microwave
data from SMMR and SMM/I used in this re-processing of ice concentrations have been extensively
used for making the ice charts in particular before the launch of wide swath SAR instruments in 1995.
In addition to the satellite data ice charts are based on information from ships and aircraft
reconnaissance. The NIC ice charts are a weekly compilation of the ice conditions. The different sea
ice categories are delineated by polygons and assigned a range of sea ice concentrations, thicknesses,
type etc. found within the polygon in the ice chart. This information is represented on the satellite pixel
grid by averaging the range of ice concentrations and other properties given within the polygon.
**2.0 Methodology**
**2.1 Atmospheric noise reduction of the brightness temperatures using NWP data**
Using an emission model, the brightness temperatures are corrected for the influence of water vapor in
the atmosphere and open water surface roughness caused by wind shear. The emission model used for
atmospheric noise reduction of the SMMR brightness temperatures, Tb, with NWP input is (Wentz,

28    1983):

$Tb = f(Ts, u*, V, L, Ta)$      (1),
where Ts is the physical surface temperature, U* is the sea surface wind friction velocity, V is the





integrated atmospheric water vapor column, L is the atmospheric liquid water column, and Ta is the
surface (at 2 m) air temperature. A similar model is used for the SSM/I data (Wentz, 1997). Over areas
with both ice and water the influence of open water roughness on the brightness temperatures and the
ice emissivity is scaled linearly with the ice concentration. The emissivity of ice is given by standard
tie-point emissivities and the total ice concentration is solved by iteration with a first guess of the ice
concentration from the NASA Team algorithm (Cavalieri et al., 1984) with static tie-points. The
correction procedure is described in detail in Andersen et al. (2006B). The NWP model grid points are
co-located with the satellite swath data in time and space using linear interpolation and a correction to
the brightness temperatures using Eq. 1 is applied. The potential inconsistencies between the ERA40
and the operational ECWMF models are minimized by the dynamical tie-point adjustment later in the
processing and eventually the residual error is included in the error estimate.
The representation of atmospheric liquid water column in the NWP data is not suitable to use for
brightness temperature correction. The data are therefore not corrected for the influence of atmospheric
liquid water. Assuming a neutral atmospheric temperature profile, the wind speed at 10 m, given by the
numerical weather prediction model, is converted to the surface friction velocity using the factor 0.047.
The other NWP variables are used directly.
**2.2 Dynamical tie-points**
Tie-points are typical signatures of ice and open water which are used in the ice concentration
algorithms as a reference. The tie-points are derived by selecting brightness temperatures from regions
of known open water and ice.
During winter, in the consolidated pack ice well away from the ice edge, the ice concentration is very
near 100 %. This has been established using high resolution SAR data, ship observations and by
comparing the estimates from different ice concentration algorithms (Andersen et al., 2007). The
apparent fluctuations in the derived ice concentration in the near 100 % ice regime are primarily
attributed to snow/ice surface emissivity and temperature and atmospheric variability around the tie-
point signature and only secondarily to actual ice concentration fluctuations. In the marginal ice zone at
intermediate ice concentrations and over open water the atmospheric emission and wind shear and
smearing dominates as error sources. There is no explicit correction for cloud liquid water and this is



an uncertainty source over both ice and open water. The fluctuations due to atmospheric and surface
emission are systematic. In fact, different algorithms with different sensitivity to atmospheric and
surface emission compute very different trends in sea ice extent on seasonal and decadal time scales
(Andersen et al., 2007). This means that not only does the estimated sea ice extent have a climatic
trend; also the atmospheric and surface constituents affecting the microwave emission are changing. In
an attempt to compensate for the influence of these artificial trends the tie-points are derived
dynamically using a window of width ± 15 days centered at the day of the actual sea ice concentration
retrieval. It is assumed that ice concentrations greater than 95 % from the NASA Team algorithm
(Cavalieri et al., 1984) are in fact a representation of near 100 % ice. The NASA Team algorithm has
different sensitivities to artificial trends than the two algorithms (see section 2.3 below) used in
combination here (Andersen et al., 2007). The ice tie-point is the mean value of these selected data
points. The static NASA Team tie-points for SMMR are found in Gloersen et al. (1992) and for SSM/I
the tie-points are found in Andersen (1998). Geographically, the sea ice tie-point is excluding data of
both the SMMR and the SSM/I instruments pole-wards of 84° for consistency between the SMMR and
SSM/I periods. The open water tie-point data were selected geographically along two belts on the
northern and southern hemisphere respectively (between 53°N and 75°N and between 65°S and 80°S).
A land mask including the coastal zone and sea ice maximum extent climatology ensures open water
data only.
There is no attempt to compensate explicitly for sensor drift or inter-sensor calibration differences
(even though the SSM/I data have been inter-calibrated) or possible biases in the NWP fields used for
atmospheric noise reduction of the brightness temperatures. The dynamical tie-point method is in
principle compensating for these problems in a consistent manner.
**2.3 The ice concentration algorithm**
The analysis of atmospheric sensitivity in Andersen et al. (2006B) showed that the Bootstrap frequency
mode algorithm (Comiso, 1986; Comiso et al., 1997) had the lowest sensitivity to atmospheric noise at
low ice concentrations. Furthermore, the comparison to high resolution SAR imagery in Andersen et al.
(2007) indicated that among the algorithms using 19 and 37 GHz channels available on both SMMR
and SSM/I - SSMIS, the Bristol algorithm (Smith, 1996) had the lowest sensitivity to ice surface





emissivity variability. In addition the Bristol algorithm had low sensitivity to atmospheric emission in
particular at high ice concentrations.
Consequently, we use a combination of the Bristol algorithm and the Bootstrap frequency mode
algorithm. The Bootstrap algorithm is used over open water and the Bristol algorithm is used over ice.
At intermediate concentrations up to 40% the ice concentration is an average weighted linearly
between the two algorithms. This hybrid algorithm is also used as the operational OSI SAF sea ice
concentration algorithm.
**2.4 The Bootstrap and Bristol sea ice concentration algorithms**
The original Bootstrap sea ice concentration algorithm is a combination of two algorithms: the
polarization mode which is used over ice and frequency mode which is used over open water (Comiso,
1986). Only the Bootstrap algorithm in frequency mode, the open water part, is used here. The
algorithm assumes only two surface types: ice and open water. The linear relationship yields the
following formulation for the total sea ice concentration, ic:
$ic_{Bootstrap} = (Tb - Tb^W \ )/(Tb^I + Tb^W),$     (2)
where $Tb$ is the measured brightness temperature, $Tb^W$ is the open water tie-point, and $Tb^I$ is the ice tie-
point.
The Bristol algorithm (Smith, 1996) is conceptually similar to the Bootstrap algorithm. In a three-
dimensional scatter plot spanned by $T_{19v}$, $T_{37v}$ and $T_{37h}$ the ice points tend to fit a plane surface. The
only difference to the Bootstrap algorithm is that instead of viewing the data in the $T_{19v}$, $T_{37v}$ space, the
Bristol algorithm views the data perpendicular to the data plane, i.e. in a transformed coordinate
system:
1. axis: $T_{37v} + 1.045T_{37h} + 0.525T_{19v},$     (3a)
2. axis: $0.9164T_{19v} - T_{37v} + 0.4965T_{37h}.$     (3b)
The remaining analysis is identical to the Bootstrap algorithm.
The Bootstrap algorithm is used over open water and the Bristol algorithm is used over ice. At





intermediate concentrations up to 40% the ice concentration is an average weighted linearly between
the two algorithms i.e.
$ic = (1 - wc) * ic_{Bristol} + wc * ic_{Bootstrap}$ (4a),
where
$wc = (|t - ic_{Bootstrap}| + t - ic_{Bootstrap})/(2 * t)$ (4b),
where t is the threshold of 40%.
**2.5 The sea ice concentration uncertainties**
The uncertainties described in the following sections are generally independent and the squared sum of
the two estimated components of uncertainty is assumed to represent the total uncertainty squared. The
tie-point uncertainty $\varepsilon_{tie\text{-}point}$, including residual atmospheric noise, sensor noise and ice surface
emissivity variability, is derived from measurements as the first component of uncertainty. The
representativeness error, $\varepsilon_{smear}$, is simulated using a model as the second component of uncertainty, i.e.
$\varepsilon_{total}^2 = \varepsilon_{tie-point}^2 + \varepsilon_{smear}^2$ (5).
**2.6 First component: instrument noise, algorithm and tie-point uncertainties**
Both the water surface and ice surface emissivity variability and emission and scattering in the
atmosphere affects the brightness temperatures and the computed ice concentrations. Different
algorithms have different sensitivities to these surface and atmospheric parameters (Andersen et al.,
2006B). Further, both the atmospheric and surface parameters affecting the ice concentration estimates
have climatic trends (Andersen et al., 2007). To reduce the uncertainties due to atmospheric noise, the
brightness temperatures are corrected using NWP data for atmospheric water vapor and open water
roughness. The dynamical tie-points reduce the uncertainty due to the climatic trends in the atmosphere
and on the ice surface on a hemispheric scale while regional trends may still exist. The remaining tie-
point uncertainties are given as the spatial ice concentration standard deviation in regions with open
water or 100% ice.
Random instrument noise also results in ice concentration uncertainties. The SSM/I instrument noise
results in an ice concentration uncertainty of 1.4 % for the Bristol algorithm, and 1.7 % for the
Bootstrap algorithm in frequency mode (Andersen et al., 2006A). Systematic sensor drift is critical
issue for ice concentration algorithms using static tie-points. Here we use dynamical tie-points intended



for alleviating problems with sensor drift, inter-sensor calibration and climatic trends in ice surface
emissivity and atmospheric emission, i.e. this method minimizes the uncertainties caused by sensor
drift.
**2.7 The geo-location error**
Geo-location error - the geo-location error occurs when the satellite is not exactly oriented. Simulations
show that because of the large footprints (see next section for footprint sizes) compared to the typical
geo-location errors (about ±5 km, Hollinger et al., 1990) the ice concentration uncertainty due to geo-
location errors is small and neglected here. Locally the geo-location errors may be significant but
difficult to estimate.
**2.8 Second component: the representativeness error**
Footprint sizes for the channels used for ice concentration mapping range from about 50-70 km for the
19 GHz channels to about 30 km for the 37 GHz channels. Footprints of uneven size are combined in
the algorithms when computing the ice concentration. The footprint ice concentration is represented on
a predefined sampling grid. The ice concentration data are normally represented on a finer grid
(typically 12.5 or 25 km) than the sensor resolution (30 to 70 km). This effect is called smearing. The
combination of footprints of uneven size in the ice concentration algorithm results in an additional
smearing effect. This we call the footprint mismatch error. The smearing and the footprint mismatch
error cannot be estimated separately. However, the combined error can be estimated if all other error
sources and the ice cover reference are known a priori. It can also be simulated using high resolution
ice concentration reference data and a model for the satellite measurement footprint patterns. Here we
use the model described in section 2.9.
**2.9 Simulating the smearing uncertainty**
The smearing simulation model uses high resolution brightness temperature input to compute the
brightness temperatures as would be measured by the coarse resolution radiometers on board the
satellite. The high resolution input is compared to the coarse resolution output and realizations of ice
concentrations in the OSI SAF hybrid algorithm.



Reference SIC is cloud-free MODIS scenes re-sampled to 1 km x 1 km pixel size described in section
1.4. The MODIS intensity may vary slightly as a function of solar angle and snow type leading to
uncertainties in the actual ice concentration. However, here it is regarded as the truth and it does
provide a realistic spatial distribution of ice at the right scale for input to the model and as a reference
for comparison. Each of these 1 km x 1 km ice concentration pixels is assigned a microwave brightness
temperature using standard tie-points (Comiso et al., 1997) and linear mixing between 0 and 100%. For
each 1 km x 1 km brightness temperature pixel elliptical Gauss-shaped antenna patterns (Drusch et al.,
1999) are used to simulate brightness temperatures at 19v and 19h, 37v and 37h as it would be
measured with SMMR and SSM/I - SSMIS on the satellite. The simulations of brightness temperatures
are used as input to the CF and Bristol algorithms using standard tie-points. The resulting ice
concentration estimate is then compared to the ice concentration reference sampled to different
resolutions, i.e. 1, 5, 10, 12, 25 and 50 km (see table 2). The STD between the truth at a certain pixel
resolution and the simulated satellite image is the smearing uncertainty. The smearing uncertainty is
assumed uniform between 0% + $\varepsilon_{tiepoint}$ and 100% - $\varepsilon_{tiepoint}$. At 0% and at 100% it logically is zero.
Table 2 shows the smearing uncertainty for the CF, the Bristol and the average OSI SAF STD of the
difference at different grid resolutions. The final grid resolution is 10 or 12 km which means that the
smearing uncertainty is 13% or 12% respectively. The smearing uncertainty is nearly the same for the
CF and the Bristol algorithms.
The MODIS image used for estimating the smearing uncertainty is shown in Figure 1. The image has
regions of open water, intermediate concentrations and of complete ice cover. The simulated SSM/I sea
ice concentration using figure 1 as input to the OSI SAF algorithm is shown in figure 2.
**2.10 The sea ice concentration uncertainty algorithm**
The representativeness uncertainty is computed as a function of ice concentration using a model. The
other error sources are computed using the hemispheric standard deviation of the measurements over
open water and over near 100% ice respectively. The ice concentration algorithm provides ice
concentrations which are greater than 100% and less than 0%. These unphysical concentrations are
truncated in the processing. Therefore, we write the ice concentration, ic:
$ic = (1 - \alpha(ic))water + \alpha(ic)ice$           (6),





where ic is the ice concentration calculated by the algorithm and $\alpha$ as a function of ic is:
$\alpha(ic) = \prod_0^1 (ic)\, ic + H(ic - 1)$     (7),
where $\prod_a^b(x)$ is the Boxcar function and H(x) the Heaviside step function.
if ic≤0 then $\alpha$=0
if 0<ic<1 then $\alpha$=ic         (8)
if ic≥1 then $\alpha$=1
Using equation 2 and assuming the uncertainty for the ice and water part is independent this leads to a
total tie-point uncertainty i.e
$\varepsilon_{tie-point}(\alpha(ic)) = \sqrt{(1 - \alpha(ic))^2 \varepsilon_{water}^2 + \alpha^2(ic)\varepsilon_{ice}^2}$     (9),
where $\varepsilon_{water} = \varepsilon(IC(P_{water}))$     (10),
and open water is determined by a monthly varying ocean mask, IC is the functional mapping of the ice
concentration algorithm and $P_{water}$ denotes the set of swath pixels for all swaths (used for calculating
the daily product).
$\varepsilon_{ice} = \varepsilon(IC(P_{NT>0.95}))$    (11),
is the STD of the ice concentrations where the NASA team (NT) algorithm finds ice concentrations
greater than 0.95.
Figure 3 shows the ice concentration uncertainty as a function of ice concentration. The total
uncertainty squared is the sum of the different uncertainties squared (see eq. 4). The smearing
uncertainty is zero for open water and for 100 % ice. At these two points on the curve there is only the
tie-point uncertainty (including sensor and residual atmospheric noise) for open water and ice
respectively. The smearing uncertainty reaches a maximum at intermediate concentrations between
(0%+$\varepsilon_{tiepoint}$) and (100% - $\varepsilon_{tiepoint}$). Uncertainty for ice concentrations smaller than 0% and greater than
100% is the tie-point uncertainty.





Because the sea ice concentration is provided on a relatively fine grid of about 10 km and 12.5 km
compared to the actual resolution of the sensor the smearing uncertainty is the component dominating
the total uncertainty. When the grid resolution is comparable to the actual spatial resolution of the
algorithm at 50 km the smearing uncertainty (see table 2) become comparable in magnitude to the tie-
point uncertainty.
**2.11 From level 2 swath projection data to level 3 daily grids to interpolated level 4 maps**
The transition from level 2 swath projection data to level 3 and 4 daily predefined EASE and polar
stereographic grids includes the gridding of the swath data, the filtering of coast line grid cells,
maximum ice extent masking and spatial and temporal interpolation.
The time window of 24 hours is centered at 12:00 UTC. The ice concentration swath data is averaged
for each grid cell using the weighting function:
$weight = 1 - 0.3 * (dist/inflrad)$   (12),
where *dist* is the distance between the data point centre and the grid cell centre and *inflrad* is the radius
of influence (18 km). All data from overlapping missions are included in the gridding except the
overlap between SMMR and SSM/I. Only the SSM/I data are used during the overlap of 1.5 months
between SMMR and SSM/I.
**2.12 Statistical filtering of ice concentration near the coastline**
Due to the coarse resolution of the radiometers the data may be influenced by land up to 50 km from
the coastline. The emissivity of land along the coastline is comparable to sea ice emissivity and much
higher than water emissivity. This means that in the coastal zone if there is open water or intermediate
concentrations the sea ice concentration will be overestimated. The statistical method which is
described in Cavalieri et al. (1999) is used for filtering the ice concentration near the coast: For each
grid cell along the coast the monthly mean and the minimum ice concentration is estimated using the
1985 SMMR and the 1992 SSM/I data. The minimum ice concentration is used instead of the estimated
ice concentration if the adjacent non-coastal grid points are ice free.





**2.13 Climatological maximum sea ice extent masking**
Occasionally spurious sea ice is detected in open water regions far from the ice edge due to
atmospheric noise affecting the ice concentration estimate. These spurious sea ice detections are
masked out using the monthly maximum extent climatology by NSIDC
(http://nsidc.org/data/smmr_ssmi_ancillary/ocean_masks.html). A zone of additional 100 km into the
open water has been added to the maximum extent to ensure detection of real sea ice outside of the
climatology.
**2.14 Level 4: Gap filling by spatial and temporal interpolation**
Only grid cells with missing data are filled with interpolated values in the level 4 processing and
interpolated values are flagged. Daily data coverage is never complete and occasionally there are
missing scan lines, missing orbits and the hole near the North Pole is never covered by the satellite.
Interpolation is efficient in filling small gaps e.g. one or two missing scan lines but it is deceiving when
large areas are missing and filled with interpolated values. To overcome this issue, yet implementing a
general approach for all cases, both temporal and spatial interpolation is used. The interpolation on a
given date D uses data from the day before and after, i.e. D-1, D and D+1.
The interpolated value at grid cell (i,j) for day $D$ is given by:
$$X_{i,j}^D = K(w_{i,j}^{D-1} X_{i,j}^{D-1} + w_{i,j}^{D+1} X_{i,j}^{D+1} + \Sigma_{k,l} W^D(k,l;i,j) X_{k,l}^D) \qquad (13),$$
where $X$ is the sea ice concentration value and $K$ is a normalizing factor given by:
$$w_{i,j}^{D-1} + w_{i,j}^{D+1} + \Sigma_{k,l} W^D(k,l;i,j) = 1/k \qquad (14).$$
The spatial interpolation from neighbors of cell (i,j) in equation 6 is only using values from date *D*,
while the temporal interpolation is only concerned with the value from the exact (i,j) cell but from
dates *D-1* and *D+1*. This ensures that the interpolation will be  efficient in the two following extreme
scenarios: 1) In a region where we never have satellite observations e.g. the data coverage gap near the
North Pole, the spatial interpolation term will be the only contribution. 2) Conversely, in the case of
several missing swaths on day D only (nominal coverage on D-1 and D+1), the interpolated values will
be computed from the previous and next days, taking advantage of the persistence of sea ice
concentration over relatively short periods. The interpolation for intermediate cases (when both spatial





and temporal neighbors exist) is a compromise of those extreme situations.
In Eq. 6, the weighting parameters are computed as follows:
$w_{i,j}^{D} = 1/(\sigma_{i,j}^{D})^2 (2N_{max} + 1)$                              (15)
$W^{D}(k,l;i,j) = 1/(\sigma_{k,l}^{D})^2 \times exp(-0.5(\frac{\Delta(k,l;i,j)}{R_{i,j}})^2)$          (16),
where σ is the standard deviation associated to each ice concentration estimate, Δ is the distance
between a given (k,l) neighbor and cell (i,j) and $R$ is an auto-correlation radius. The spatial
interpolation weight is thus based on an isotropic Gaussian distribution, and almost all (>99.9%) of the
interpolation weight is concentrated inside a [-3R;+3R] x [-3R;+3R] km$^2$ area, which translates into a [-
$N_{max}$;+$N_{max}$] x [-$N_{max}$;+$N_{max}$] grid cells squared area. It was found by testing that $R$ is proportional to the
absolute latitude in degrees, i.e. $R$ = latitude of (i,j).
For the SMMR which was operated every second day, the temporal interpolation is *D-2* and *D+2*
instead of *D-1* and *D+1* for SSM/I.
**3. Results and discussion**
We compared the ESICR to sea ice charts for reference during the period from 1978 to 2009 on both
hemispheres. There is a gap in the comparison on the southern hemisphere because we did not have
access to ice charts between Dec. 1994 and Jan. 2003. The overlap period during July and August 1987
between the SMMR and the SSM/I instruments will be analyzed in more detail in section 3.2. The
latter period from 2009 to 2014 is not compared to ice charts.
It is clear that ice charts are produced to support ship and offshore operations and not to monitor sea ice
as a climate parameter. However, it is a relatively independent dataset with a long history, produced in
a relatively consistent manner therefore we use it for comparison here.
**3.1 The ice concentration comparison to sea ice charts**
The entire period from 1987 to 2009 is covered by ice charts from the NIC on the northern hemisphere.
For the southern hemisphere there is gap from Dec. 1994 to Jan. 2003. The ice charts and the sea ice
climate record are gridded onto the 12.5 km EASE grid and compared. The total concentration in the
ice chart is given as a range of sea ice concentrations describing the variability within each ice chart





polygon. For each ice chart concentration level (the total concentration) the deviation between ice chart
and the ice concentration is computed and the bias and standard deviation is calculated for each
concentration level. The bias and standard deviation are reported for ice (> 0% ice concentration), for
water (0% ice concentration) and for both ice and water as a total.
The bias in ice concentration between the Northern Hemisphere National Ice Center ice charts and
ESICR ice concentration is shown in figure 4. The ESICR ice concentration is higher than the ice chart
over open water. This is due to the fact that the radiometer ice concentration is affected by atmospheric
noise which increases the ice concentration above zero. The ice charts have a nominal value of zero
over open water. Actually the mean open water ESICR ice concentration is zero. However, all negative
ice concentration estimates are truncated to zero which leaves the small positive bias. Also the
uncorrected noise from cloud liquid water, water vapor and wind over open water gives a positive bias
in the ESICR ice concentrations. This positive bias is not present in the ice charts. The SMMR to
SSM/I transition in 1987 is seen as a small increase in the open water bias because the SSM/I 19.35
GHz is affected more by water vapor than the 18.0 GHz SMMR instrument. Apparently not all the
noise due to water vapor in the atmosphere is removed successfully in the atmospheric correction
scheme. However, the higher noise level in SSM/I is quantified in the uncertainties. The ice bias has a
clear seasonal cycle and a negative winter bias around 5% to 10%. The negative bias is caused by the
truncation of the over 100% ice concentrations. The negative summer sea ice bias is sometimes
reaching 20%. This is caused by anomalous sea ice emissivities during melt, the presence of melt-
ponds, and perhaps an overestimation of the ice concentrations in the ice chart.
Figure 5 shows the northern hemisphere standard deviation of the difference between the ESICR and
the national ice center ice charts. Both the standard deviation of open water and ice has a clear seasonal
cycle with higher standard deviations during summer than during winter. The standard deviation of
open water is decreasing during the latter part of the record. This could be a result of higher quality
wind and water vapor data in the recent part of the ERA40 reanalysis and in the operational ECWMF
model used since 2002.
Figure 6 shows the ESICR and national ice center difference for ice, water and both ice and water for





the southern hemisphere. There were no digital ice charts available between Dec. 1994 and Jan. 2003.
There is a small positive bias over open water due to the truncation of spurious sub-zero ice
concentrations in the ESICR. The near 100% sea ice ESICR and NIC chart difference is negative
around -10% during Antarctic winter. During the Antarctic summer the difference over ice is near -

5   20%.

The standard deviation of the difference between the ESICR and the NIC ice charts shown in figure 7
is higher and has more inter-annual variability in Antarctica than in the Arctic except for the open
water. The standard deviation of the difference for the open water case is between 0 and 5% from 2003
onwards.
**3.2 The SMMR and SSM/I overlap**
The overlap period between SMMR and SSM/I during July and August 1987 is short because 15 days
prior and after the actual date is needed in order to establish the tie-points properly. Subtracting 15 days
in each end of the overlap period leaves only a few days where the tie-points are fully established.
Figure 8 and 9 show the overall bias between SMMR and SSM/I including the periods where tie-points
are based on less than one month data, for NH and SH, respectively. On the Northern Hemisphere the
overlap is during the sea ice minimum in 1987 which means that there are a limited number of ice data
points.
The SMMR and SSM/I overlap period coincides with the ice maximum on the Southern Hemisphere
which is ideal for comparison. However, the comparison is limited by the very short overlap just as for
the Northern Hemisphere. Inspecting the differences geographically indicates that when environmental
conditions have not changed significantly during SMMR and SSM/I passes then the SSM/I is slightly
higher over open water while over ice it is close to neutral. The open water bias is probably due to the
higher sensitivity of the 19.35 GHz channel on SSM/I to water vapor than the 18.0 GHz channel on
SMMR.
**3.3 Ice chart and ESICR comparison discussion**
The NIC ice charts are produced manually on the basis of satellite and reconnaissance data for ship



navigation support and they do not contain estimates of uncertainty. However, the uncertainties in the
ice charts from two other producers are relatively large with standard deviation of the difference
between overlapping and coincident Greenland and Norwegian ice charts up to 30% ice concentration
especially at intermediate concentrations (Breivik et al., 2015). Compared to microwave radiometer ice
concentrations (the OSI SAF operational algorithm in Andersen et al., (2006B)) the ice concentration
in Greenland ice charts is about 30% higher at intermediate concentrations. Trials with the ice
concentration model described in section 2.9 shows that the estimates from most sea ice concentration
algorithms including the Bootstrap and the Bristol is 1:1 proportional to the actual ice concentration
and that there are very small differences between the overall response of different algorithms (ice
concentration differences < 1% on 1000 km scale not including noise), i.e. different algorithms yield
same ice concentrations given the same brightness temperature input. We did not find a similar
investigation comparing NIC and other overlapping and coincident ice charts. However, we note that
the methodology for making the Greenland, Norwegian and NIC ice charts is similar.
The bias between ice charts and radiometer ice concentrations at intermediate concentrations could be
caused by two effects: 1) estimated radiometer ice concentrations are lower than real ice concentration
for new ice and if the surface is melting or refrozen after melting and both new ice and melting
refreezing is abundant in regions with intermediate concentrations, i.e. the radiometer is
underestimating the real ice concentration 2) The ice charts ice concentration is a subjective estimate
which is made for the safety of navigation and the overestimation of the ice concentration in the ice
chart stem from "better-safe-than-sorry" practices within the ice charting community.
**3.4 The ESICR metrics**
In the following we are giving examples of the ESICR dataset for estimating sea ice climate statistics
and trends. The applied climate period here is the full length of ESICR. First we show the long term
trend in sea ice extent and secondly the trend in open water days in regions covered part of the season
by sea ice. The examples are given for both the northern and the southern hemisphere.
Here the sea ice extent is defined as the area covered by sea ice within the ice edge. The ice edge is
defined as the 30% contour and ice concentrations greater than 30% are considered as ice covered



while concentrations less than 30% are considered open water.
For the Arctic there is a negative trend in the monthly mean extent for all months of the year. The
negative slope is largest in September -94 000±9700 km²/yr and smallest in May -32 000±4600 km²/yr.
The monthly trends for the Arctic are shown in Table 3A.
For the Antarctic there is a positive trend in the monthly mean extent for all months of the year. The
positive slope is largest in April, October and December at 33 000 km²/yr and the smallest in February
13 000±5400 km²/yr. The monthly trends for the Antarctica are shown in Table 3B.
Below we have looked at two periods of the 35 year ESICR: the entire 35 year period from autumn
1978 to the end of 2014 and the shorter recent 10 year period from 2004 to the end of 2014. The latter
shorter period represents the period where most of the sea ice extent changes are taking place in both
the southern and northern hemisphere.
Figure 10 shows the sea ice extent for the Arctic for both the long and the short record together with
the September 2012 sea ice extent. The lower two panels are showing the seasonal variability of the sea
ice extent and the long term mean monthly sea ice extent in March and in September which is the
maximum and minimum extent respectively. September 2012 was the lowest sea ice extent on record
in the Arctic since beginning of the satellite era. Over the 35 years of ESICR there is a negative trend in
sea ice extent for all months of the year with the largest negative trend during the summer and the
beginning of autumn (Jul-Oct) i.e. the third quarter of the year (Q3).
Figure 11 shows the sea ice extent for the Antarctic for both the long and the short record together with
the September 2012 sea ice extent. The lower two panels are showing the seasonal variability of the sea
ice extent and the long term mean monthly sea ice extent in March and in September which is the
minimum and maximum extent respectively. The sea ice extent has experienced an overall positive
trend around Antarctica especially downstream of the Weddell and the Ross Seas in the clockwise
atmospheric circulation along the ice edge.





In order to determine the period of open waters for a given pixel, the annual spatial distribution of dates
of freeze-up and break-up were calculated. The freeze-up date for a given point is defined as the date
where the sea ice concentration climbs from below to above 30% and remains so for at least 5 days.
The break-up date for a given point is defined as the date where the sea ice concentration falls from
above to below 30% and remains so for at least 5 days.
The values for the ice concentration threshold and length of period were chosen by manually tuning for
convergence: ice concentrations lower than 30% and periods less than 5 days were found to produce
noise in the spatial distribution of freeze-up/break-up dates, which settles at the chosen values, though
somewhat less so in the short 10 year record.
Since the sea ice does not retreat and expand completely every year, not all areas experience the same
number of freeze-ups and break-ups over an equal period of years. Therefore, some regions may
experience relatively few freeze-ups and break-ups, thus reducing the confidence in the trend of the
region. As a consequence, only areas having experienced more than 6 freeze-ups/break-ups in each
period are considered.
Figure 12 is showing the decadal trend in open water days in the Arctic region covered by sea ice part
of the year. The open water days are calculated as the difference in days between freeze-up and break-
up.
The trends in open water days are shown for both the long and the short climate record in figure 12 left
and right, respectively.
Over the long record of 35 years the ice season has been shortened by at least 60 days in the Davis
Strait and in large parts of the Barents Sea. The ice season has been shortened consistently all over the
Arctic except in the Bering Strait region and the Greenland Sea. The shortening of the ice season is due
both to a delay of the freeze-up and earlier breakup in combination (not shown). While this pattern is
largely consistent for the short and the long periods in the Baffin Bay, and the Barents, Kara and
Laptev Seas there are large differences in open water days trend in the Davis Strait and in the Beaufort



Sea and Bering Strait region. The short period has substantial negative trends in these regions (more
than -15 days / decade) while the long period has positive trends. However, the statistical significance
of the trends for the short period is lower than for the long period.
Figure 13 is showing the significance of the trends in number of open water days shown in figure 13
here as a test of the null-hypothesis, i.e. testing the probability of no trend. This means that a low
probability indicates that the trend is in fact significant. It is noted that while the trend is significant in
most regions for the long record the trends are not significant for the short record. This is due to the
relatively short record of 10 years which is influenced by short term natural variability for example
shifts in the mean location of the atmospheric pressure systems.
Figure 14 shows the decadal trend in the number of open water days around Antarctica in regions with
a seasonal sea ice cover. As for the Arctic the open water days is calculated as the difference in days
between freeze-up and breakup.
The trend in open water days is shown for both the long and the short record.
The significance of the trends in number of open water days per year is shown in figure 15 as a test of
the null-hypothesis, i.e. testing the probability of no trend. It is noted that while the trend is significant
in large regions in the Weddell Sea and in the Ross Sea for the long record the trends are more
sporadically significant for the short record.
The negative trend on the long record in the Ross and in the Weddell Seas indicates that the ice is
staying longer in these areas now than before. Along the ice edge in the Ross Sea, in East Antarctica,
the Weddell Sea and in all of the Bellinghausen Sea there is a positive trend in the number of open
water days. This means that the ice which is either advected into or formed in these regions is staying
there for shorter time now than before and it indicates that these regions have experienced warming
during the 35 years of the record. Even though there is an overall positive trend in the sea ice extent
around Antarctica there is an indication that the warming is closing in on the pole.



**4.0 Conclusions**
A sea ice climate record covering the period from autumn 1978 to the end of 2014 has been produced
based on past microwave radiometer data from SMMR, SSM/I and SSMIS. The climate record has
been produced according to 4 principles to ensure consistency and to minimize the sensitivity to noise
sources:
1) Finding algorithms with low sensitivities to geophysical noise. Two algorithms have been selected in
combination based on the evaluation in Andersen et al., (2007), the Bristol over ice and the Bootstrap
in frequency mode over open water. An independent evaluation of algorithms in Ivanova et al. (2015)
pointed at the same two algorithms.
2) Regional error reduction correcting the brightness temperatures for water vapor in the atmosphere
and wind over open water. The scheme described in Andersen et al. (2006B) is used to reduce the noise
over both ice and water.
3) Calibrate the algorithms to the actual ice and water signatures and sensor drift using dynamical tie-
points. The result of using dynamical tie-points has been demonstrated here at the transition from
SMMR to SSM/I with satisfactory results. In addition, we do not see any jumps at sensor transitions or
long term trends in the comparison to the independent ice chart dataset.
4) Quantify the residual uncertainties. A forward model for the residual uncertainties has been
developed and applied. The total uncertainty as a combination of the tie-point variability and the
representativeness uncertainty is a function of the ice concentration and it is applied on each individual
measurement.
It is clear that the sea ice covers on both hemispheres have undergone large changes over the 35 year
period. In the Arctic the linear trend at sea ice minimum month in September is -94 000 km²/yr.
Around Antarctica there has been an increase of the total sea ice extent during all months especially
downstream of the Weddell Sea and in the Ross Seas. However, these extensions are relatively short



lived meaning that the ice which is extending across the long term mean extent (primarily driven by

advection) near sea ice extent maximum into the Atlantic and the Pacific ocean is removed by melt or

advection relatively quickly. However, there are regional differences and the ice extent has decreased

along the Antarctic Peninsula in the Bellinghausen Sea.

### 4.1 Future work

The sea ice climate record will be updated at irregular intervals. The next update is planned for autumn

2016. In addition, the daily OSI SAF sea ice concentration product and the ESICR is using the same

algorithm and methodology with only minor differences.

In order to extend the sea ice climate record with past data it is being investigated if it is possible to

retrieve the Nimbus 5 Electrically Scanning Microwave Radiometer (ESMR) 19 GHz swath data from

1972 to 1977. These single channel data are significantly different from SMMR and SSM/I - SSMIS

data and a new sea ice algorithm would have to be used.

The next update will include development from the ESA sea ice climate change initiative project

working towards improved sea ice climate record methodologies.

**Acknowledgements**

The work was completed with support from EUMETSAT's Ocean and Sea Ice Satellite Application

Facility. The SMMR data were provided by the NSIDC, the SSM/I data by Remote Sensing Systems,

and the numerical weather prediction model data by the ECMWF.

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





**Tables**
Table 1. The different satellite missions carrying the SMMR, SSM/I and SSMIS instrument and the
periods they cover.
Table 2. The STD of the difference between the simulated SSM/I - SSMIS satellite ice concentration
and the reference ice concentration resampled to different grid resolutions in percent.
Table 3A. The mean monthly sea ice extent, long term trend and standard error of the trend in the
Arctic. All figures are in millions of km².
Table 3B. The mean monthly sea ice extent, long term trend and standard error of the trend in the
Arctic. All figures are in millions of km².

| Sensor | Launch | End |
|---|---|---|
| Nimbus 7 SMMR | October 1978 | August 1987 |
| DMSP F8 SSM/I | June 1987 | December 1991 |
| DMSP F10 SSM/I | December 1990 | November 1997 |
| DMSP F11 SSM/I | November 1991 | May 2000 |
| DMSP F13 SSM/I | March 1995 | November 2009 |
| DMSP F14 SSM/I | May 1997 | August 2008 |
| DMSP F15 SSM/I | December 1999 | - |
| DMSP F16 SSMIS | October 2003 | - |
| DMSP F17 SSMIS | November 2006 | - |
| DMSP F18 SSMIS | October 2009 | - |
| DMSP F19 SSMIS | April 2014 | - |

Table 1. The different satellite missions carrying the SMMR, SSM/I and SSMIS instrument and the
periods they cover.

|  | 1 km | 5 km | 10 km | 12 km | 25 km | 50 km |
|---|---|---|---|---|---|---|
| CF | 18 | 16 | 14 | 13 | 10 | 7 |
| Bristol | 17 | 15 | 13 | 12 | 10 | 6 |
| OSISAF | 17 | 15 | 13 | 12 | 9 | 6 |

Table 2. The STD of the difference between the simulated SSM/I - SSMIS satellite ice concentration
and the reference ice concentration resampled to different grid resolutions in percent.




| Month | Mean [$10^6$ km²] | Trend [$10^6$ km²/yr] | Trend std err |
|-------|-------------------|------------------------|----------------|
| Jan | 14.641 | -0.045 | 0.0040 |
| Feb | 15.505 | -0.045 | 0.0043 |
| Mar | 15.620 | -0.041 | 0.0042 |
| Apr | 14.772 | -0.036 | 0.0048 |
| May | 13.403 | -0.032 | 0.0046 |
| Jun | 11.899 | -0.053 | 0.0044 |
| Jul | 09.667 | -0.079 | 0.0060 |
| Aug | 07.458 | -0.084 | 0.0083 |
| Sep | 06.881 | -0.094 | 0.0097 |
| Oct | 09.053 | -0.077 | 0.0089 |
| Nov | 11.138 | -0.055 | 0.0052 |
| Dec | 13.241 | -0.044 | 0.0043 |

Table 3A. The mean monthly sea ice extent, long term trend and standard error of the trend in the
Arctic. All figures are in millions of km².

| Month | Mean [$10^6$ km²] | Trend [$10^6$ km²/yr] | Trend std err |
|-------|-------------------|------------------------|----------------|
| Jan | 04.566 | 0.022 | 0.0092 |
| Feb | 02.911 | 0.013 | 0.0054 |
| Mar | 04.105 | 0.022 | 0.0072 |
| Apr | 06.860 | 0.033 | 0.0099 |
| May | 10.135 | 0.032 | 0.0089 |
| Jun | 13.229 | 0.029 | 0.0072 |
| Jul | 15.622 | 0.022 | 0.0055 |
| Aug | 17.129 | 0.022 | 0.0059 |
| Sep | 17.684 | 0.029 | 0.0089 |
| Oct | 17.278 | 0.033 | 0.0070 |
| Nov | 15.164 | 0.020 | 0.0065 |
| Dec | 09.932 | 0.033 | 0.0115 |

Table 3B. The mean monthly sea ice extent, long term trend and standard error of the trend in the
Arctic. All figures are in millions of km².





**Figures**
Captions:
Figure 1. The 1 km cloud free MODIS image 3000 x 2200 km. The scene is situated north of McMurdo
Station and east of the Ross Sea, Antarctica. Ice concentrations between 0% (black) and 100% (white).
The scene is recorded at 03.30 UTC 2008/02/24 by the Aqua satellite. The scene center is at 69.5S,
165W.
Figure 2. The simulated ice concentrations using the SSM/I sensor specifications and the OSI SAF
hybrid ice concentration algorithm and the data in figure 1 as input. Ice concentrations between 0%
(black) and 100% (white).
Figure 3. The total uncertainty in blue and its two components the smear in red and the tie-point
uncertainty in green as a function of ice concentration.
Figure 4. The Arctic ESICR - NIC ice chart difference for areas of ice in red, for areas of open water in
black and the total, i.e. both ice and water, in blue.
Figure 5. The Arctic ESICR - NIC ice chart standard deviation of the difference for areas of ice in red,
for areas of open water in black and the total, i.e. both ice and water, in blue.
Figure 6. The Antarctic ESICR - NIC ice chart difference for areas of ice in red, for areas of open water
in black and the total, i.e. both ice and water, in blue.
Figure 7. The ESICR and NIC ice chart standard deviation of the difference around Antarctica. The
blue curve is showing the total standard deviation of the difference for both areas of open water and
ice. The red curve is for ice and the black curve is for water. No ice charts were available to us from
1994 to 2003.
Figure 8. The overlapping SMMR - SSM/I difference in the Arctic during summer melt. The blue
curve is the total bias and the red curve is showing the ice bias.



Figure 9. The overlapping SMMR - SSM/I difference around Antarctica during austral winter. The blue
curve is the total bias and the red curve is showing the ice bias.
Figure 10. The upper panel: the September 2012 sea ice extent in the Arctic compared to the mean
extent for the long (left) and the short record (right) shown with the red line. The blue lines on either
side of the mean extent line are the 5 and 95 percentiles of ice extent. The lower two panels are
showing the annual cycle of sea ice extent. The shaded areas are the 5 and 95% percentiles. The lower
panel is showing the long term (1978-2014) Arctic sea ice extent near its maximum in March and near
its minimum in September.
Figure 11. The upper panel: the September 2012 sea ice extent in the Antarctic compared to the mean
extent for the long and the short record shown with the red line. The blue lines on either side of the
mean extent line are the 5 and 95 percentiles of ice extent. The lower two panels are showing the
annual cycle of sea ice extent. The shaded areas are the 5 and 95% percentiles. The lower panel is
showing the long term (1978-2014) Antarctic sea ice extent near its maximum in March and near its
minimum in September.
Figure 12. Show the linear trend in open water days in the Arctic for the long record (1978-2014) to the
left and the short record (2004-2014) to the right.
Figure 13. The probability that the trend in figure 12 is not significant (test of the null-hypothesis). A
low value (< 5) indicates that the trend is significant.
Figure 14. Show the linear trend in open water days in the Antarctic for the long record (1978-2014) to
the left and the short record (2004-2014) to the right.
Figure 15. The probability that the trend in figure 14 is not significant (test of the null-hypothesis). A
low value (< 5%) indicates that the trend is in fact significant.





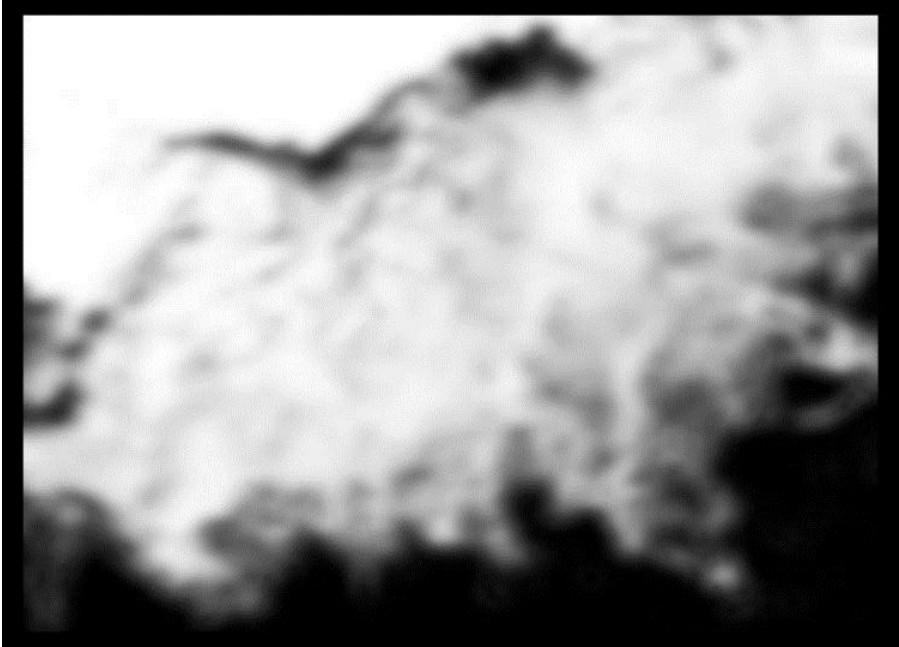

Figure 1. The 1 km cloud free MODIS image 3000 x 2200 km. The scene is situated north of McMurdo
Station and east of the Ross Sea, Antarctica. Ice concentrations between 0% (black) and 100% (white).
The scene is recorded at 03.30 UTC 2008/02/24 by the Aqua satellite. The scene centre is at 69.5S,
165W.





Figure 2. The simulated ice concentrations using the SSM/I sensor specifications and the OSI SAF
hybrid ice concentration algorithm and the data in figure 1 as input. Ice concentrations between 0%
(black) and 100% (white).

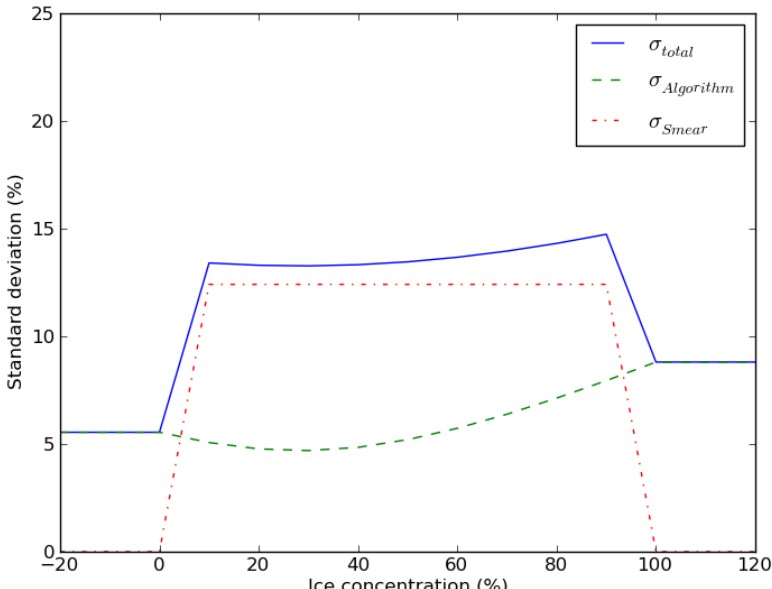

Figure 3. The total uncertainty in blue and its two components the smear in red and the tie-point
uncertainty in green as a function of ice concentration.

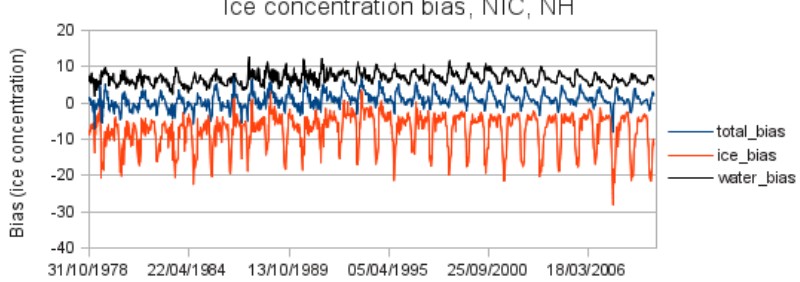

Figure 4. The Arctic ESICR - NIC ice chart difference for areas of ice in red, for areas of open water in
black and the total, i.e. both ice and water, in blue.





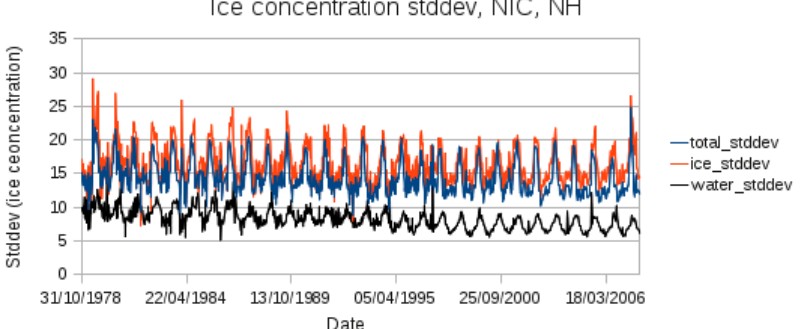

Figure 5. The Arctic ESICR - NIC ice chart standard deviation of the difference for areas of ice in red,
for areas of open water in black and the total, i.e. both ice and water, in blue.

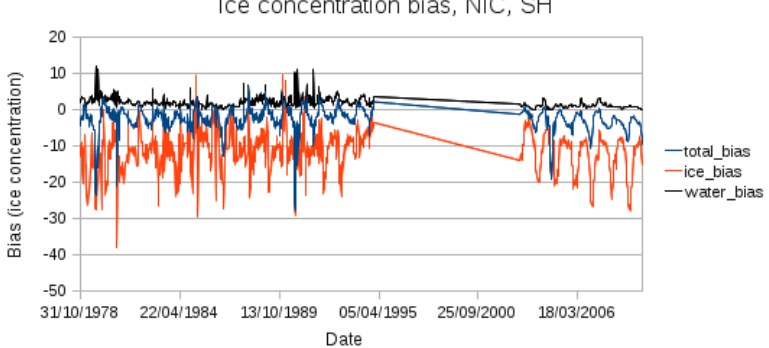

Figure 6. The Antarctic ESICR - NIC ice chart difference for areas of ice in red, for areas of open water
in black and the total, i.e. both ice and water, in blue.

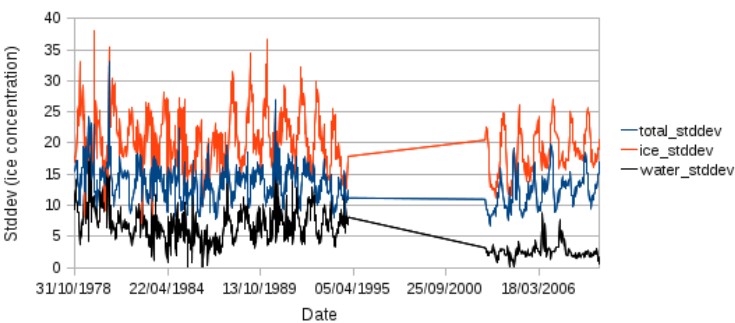

Figure 7. The ESICR and NIC ice chart standard deviation of the difference around Antarctica. The
blue curve is showing the total standard deviation of the difference for both areas of open water and
ice. The red curve is for ice and the black curve is for water. No ice charts were available to us from
1994 to 2003.



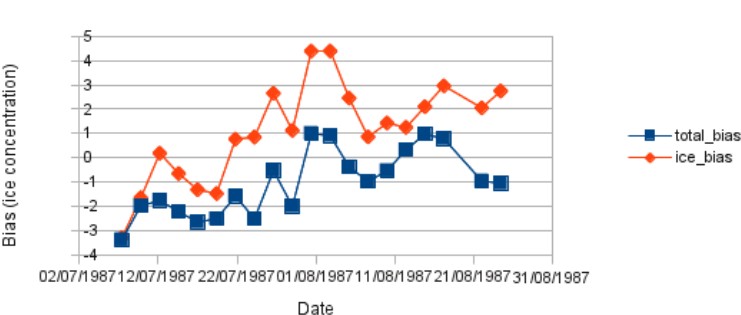

Figure 8. The overlapping SMMR - SSM/I difference in the Arctic during summer melt. The blue
curve is the total bias and the red curve is showing the ice bias.

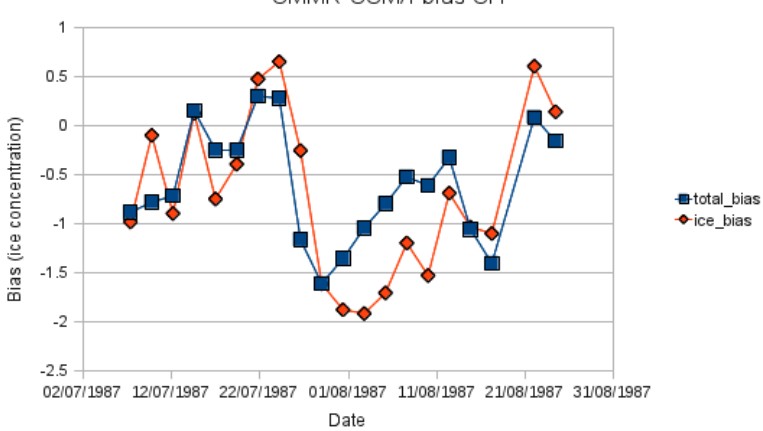

Figure 9. The overlapping SMMR - SSM/I difference around Antarctica during austral winter. The blue
curve is the total bias and the red curve is showing the ice bias.





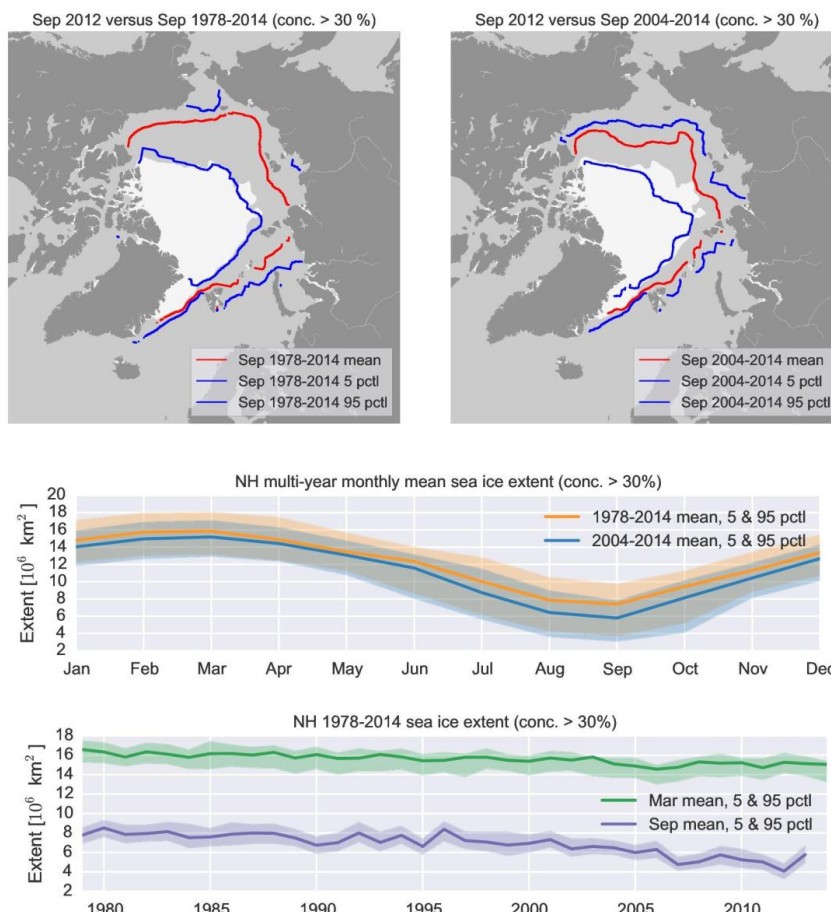

Figure 10. The upper panel: the September 2012 sea ice extent in the Arctic compared to the mean
extent for the long (left) and the short record (right) shown with the red line. The blue lines on either
side of the mean extent line are the 5 and 95 percentiles of ice extent. The lower two panels are
showing the annual cycle of sea ice extent. The shaded areas are the 5 and 95% percentiles. The lower
panel is showing the long term (1978-2014) Arctic sea ice extent near its maximum in March and near
its minimum in September.



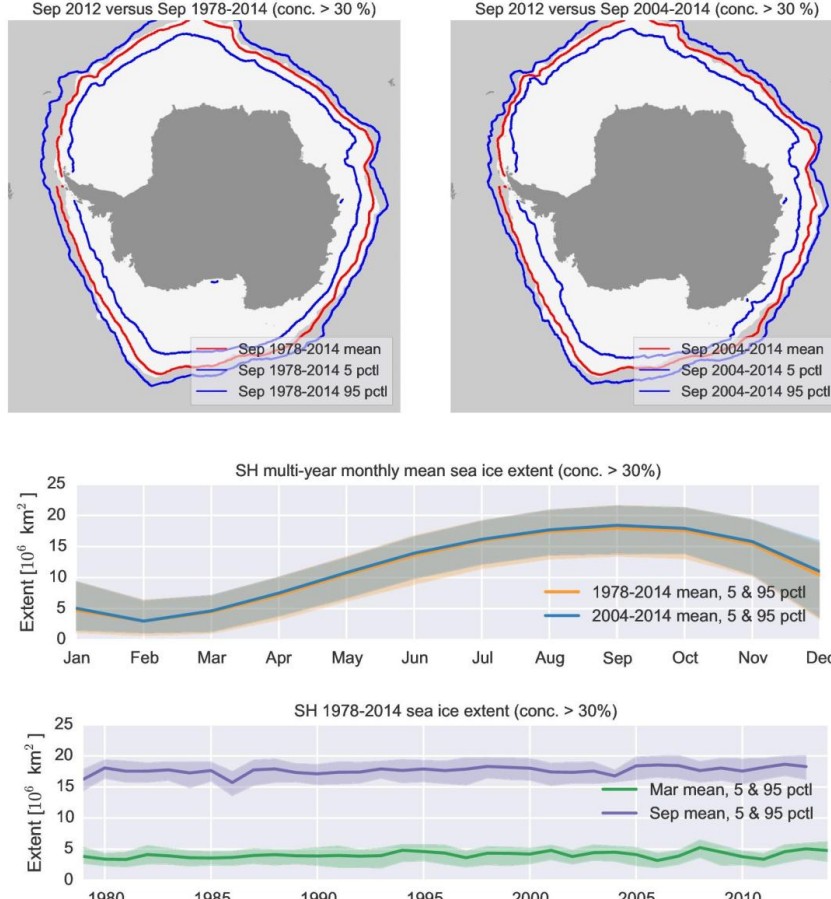

Figure 11. The upper panel: the September 2012 sea ice extent in the Antarctic compared to the mean
extent for the long and the short record shown with the red line. The blue lines on either side of the
mean extent line are the 5 and 95 percentiles of ice extent. The lower two panels are showing the
annual cycle of sea ice extent. The shaded areas are the 5 and 95% percentiles. The lower panel is
showing the long term (1978-2014) Antarctic sea ice extent near its maximum in March and near its
minimum in September.



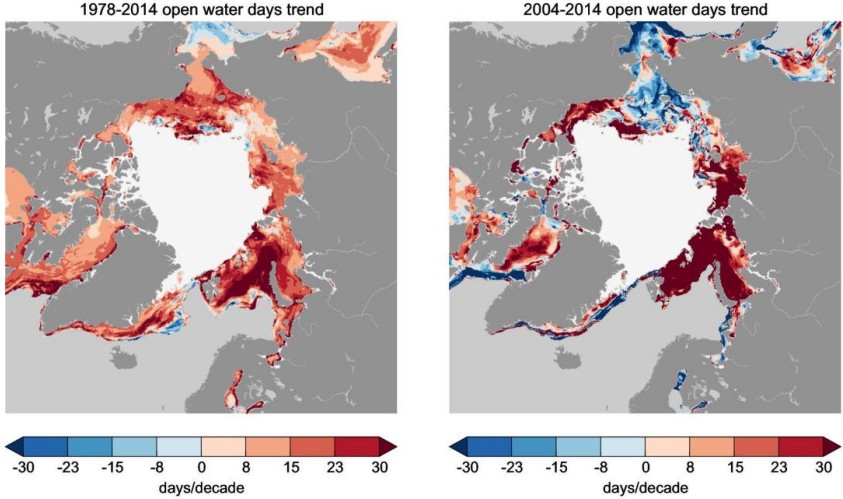

Figure 12. Show the linear trend in open water days in the Arctic for the long record (1978-2014) to the left and the short record (2004-2014) to the right.

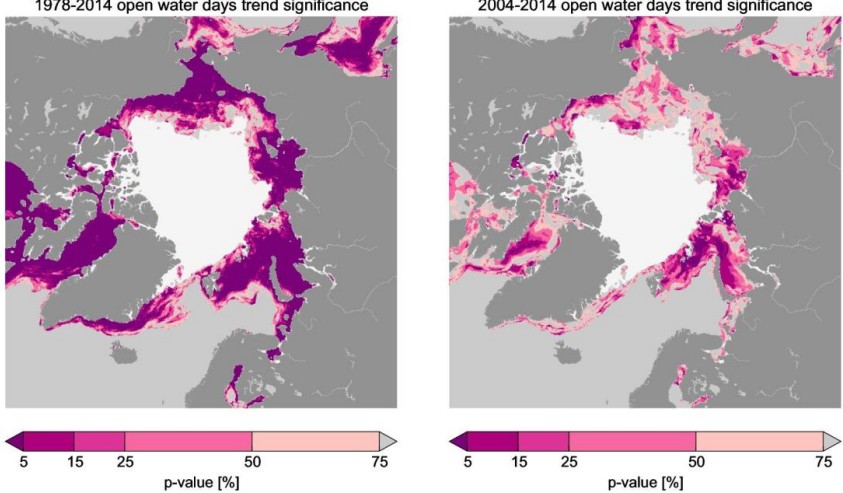

Figure 13. The probability that the trend in figure 12 is not significant (test of the null-hypothesis). A low value (< 5) indicates that the trend is significant.


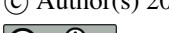

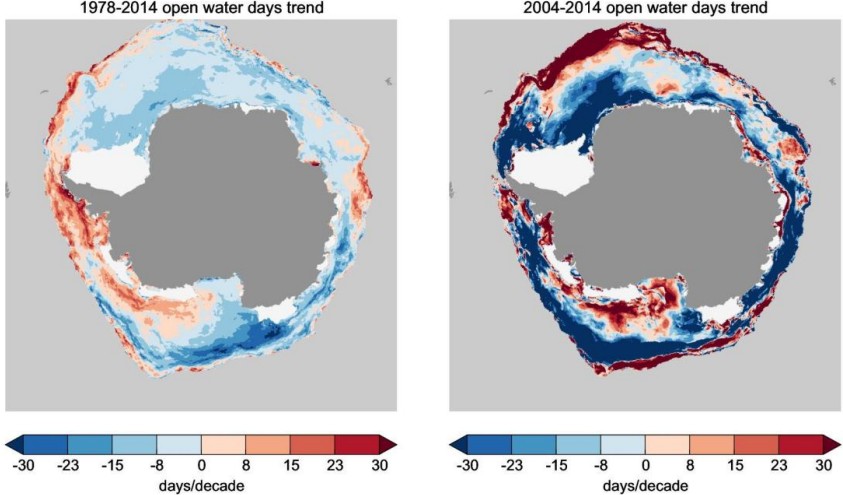

Figure 14. Show the linear trend in open water days in the Antarctic for the long record (1978-2014) to
the left and the short record (2004-2014) to the right.

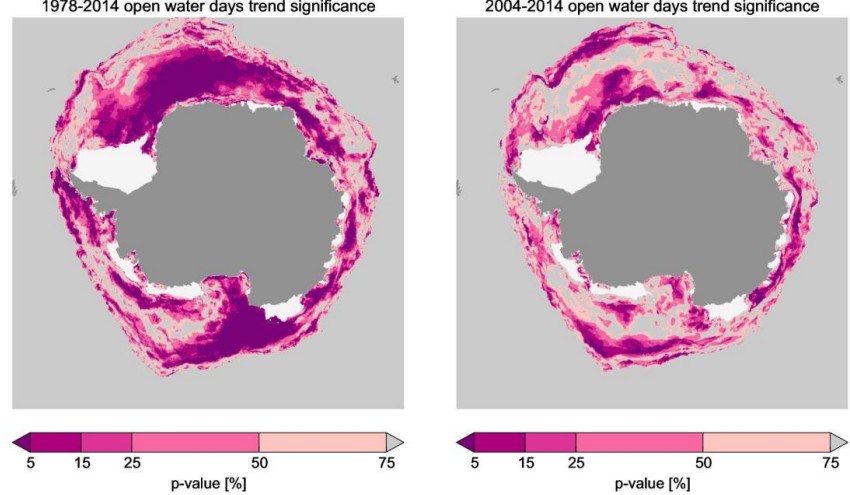

Figure 15. The probability that the trend in figure 14 is not significant (test of the null-hypothesis). A
low value (< 5%) indicates that the trend is in fact significant.

