# Peer review of "The EUMETSAT sea ice concentration climate data"

_The Cryosphere, 2016_

## Referee Comment (RC1) · I. Rubinstein (Referee) · 13 Apr 2016

General comments: The content of this publication can be considered very important contribution to the creation of consistent several decades span of sea ice cover data sets. This article reads like it was written by several different people. Up to the section 2.1 the writing style and terminology used do not explain clearly the data sets properties used in this project. The list of recommendations below will improve the readability of these sections. Starting with section 2.1 the writing style and the clarity greatly improves and very few minor clarifications are needed. But, very important section 3.3 Ice chart and ESICR comparison discussion is not easy to read. The comparison should be better organized. The authors should spend some time on deciding what will be the primary use of these long term data sets. The spatial and temporal resolutions, as well as the accuracy of retrievals are usually driven by these applications. The criteria for comparisons with existing data sets and/or ice charts are also dictated by who will

be using this new and improved sea ice cover information. Algorithms designed for processing spaceborne measurements are developed for specific applications. Some of these algorithms are very simple and set up to provide yes/no indicators to mask ice covered (>threshold value) areas. Some other algorithms, especially to be used in near-real time for navigational support, are more complex and are region and season specific. The regional and seasonal variability of sea ice cover can be partially compensated by using seasonal, regional algorithm parameters and sea ice signatures. These parameters can be derived using RTM. The complexity of the microwave radiation from the sea ice cover must be accommodated. The authors do state that understanding what contributes to the emission from the ocean surface (e.g., snow free ice floes, ice flows covered heavy wet snow, young saline ice, etc..etc„) is very important but do not carry out their evaluations region by region. The "new" hybrid algorithm should have region specific "flagging" to allow global processing. The authors should dedicate more discussion time in planning their future work to regional seasonal variability of the ice cover and how that impacts microwave and optical brightness temperatures. The authors have selected a hybrid algorithms that uses fixed coefficients multiplied by observed brightness temperatures. It is not clear how using NWP is incorporated into calculations of these coefficients or in computations of dynamic tie-points. All throughout descriptions of different algorithms the authors discuss items they call "uncertainties". It is not clear what this particular quantity represents. Is it a statistical value or derived value related to variability (or accuracy) in measured brightness temperatures or other measured (or derived) parameters? In a real world there are very few instruments that can claim to have accuracy of less than 2%. The sea ice cover data sets generated using MODIS and/or SAR provide higher spatial resolution data but the authors should not forget that the algorithms used to generated ice cover imagery may contain sensor/processing algorithm specific errors, e.g., ocean surface roughness effects on SAR imagery; cloud cover on MODIS The authors have acknowledged the need for the algorithms to accommodate pack ice and marginal ice regions. It is not clear if the hybrid algorithm will be tested for these diverse ice cover regions.

It is also not clear how different coefficients used in this new algorithm can be derived using physical properties of different types of ice and weather conditions.The ice cover in marginal ice zones can change very quickly depending on the wind (e.g., read about icebreakers trapped in Barents ice; ships stuck in Weddell sea) and in some parts of the world the extent of the ice cover is dictated by the thermal currents (e.g., sea ice cover in Labrador and Newfoundland waters). The authors have discussed "smearing" of ice cover location but did not address "smearing" effects of orbital data gridding onto a fixed grid. These effects can be determined by comparing retrievals from ascending and descending satellite overpasses looking at ice edge that was not changed (in 10 to 12 hours between satellite passes) by prevailing winds The authors should have more systematic plan for testing their chosen algorithm. Signature areas of predominant sea ice type cover (e.g., Sea of Okhotsk for seasonal ice cover; old ice covered areas; mid-winter Labrador Sea ice cover) are ideal for testing algorithms. .

. Section specific comments: In the abstract 1. Replace: American by NASA, NOAA, or NSIDC depending which agency was responsible for the satellite sensor, satellite launch, and satellite data processing Sea ice area by sea ice covered area 2. Sentence starting on line 16: "The methodology.........and ending on line 21 is not clear 3. Are the authors saying that in preparation of this sea ice dataset : 1) NWP and RTM were used to reduce impact of the atmospheric conditions on the measured brightness temperatures; 2)to reduce the inter-sensor comparison bias dynamic tie points were used in retrieval algorithms, and 3) hybrid algorithm was created using Bristol and Bootstrap algorithms. This algorithm was used in estimation of the spatial and temporal variabilities in sea ice concentration retrieval accuracy. 4. Who generated "sea ice charts from the Arctic and the Antarctic" (lines 22 and 23) 5. What are the "intermediate ice concentration" 6. Line 32: see comment #1 7. Pg. 2, line 1: by "Here the sea..." The authors are saying In publications referenced above the sea ice..... 8. Pg. 2 line 8: replace "atmospheric parameters" by weather conditions such as wind generated ocean surface roughness and cloud cover 9. What is "structural uncertainties" 10. Line 15 Define "noise" 11. Why "climatic trends" are "artificial trends" ??? 12. Be more

specific in "for the properties that we are able to quantify,????? 13. Please explain what are the "dynamical tie-points" 14. "residual uncertainties" for which parameters and please, define uncertainty is it - accuracy.

In section 1.1 Line 14 after "84°." Should be replaced by: "SMMR data were acquired every second day" In Section 1.2 "The SSMIS data (used in this project?) are from the L2B near real time data-stream" received from??? In section 1.3: Please clarify your statement on lines 7 and 8. Was sea ice coverage included in NWP? If yes, where did that information come from? In section 1.4 line 15 insert "The coarse resolution of the passive microwave brightness..." Line 22 and line 24 "Brightness above 220 (units?) 60 (units?) "We use 6 hourly data at a resolution of 1.25 degrees." Why this temporal and spatial resolution was used?

Section 1.4 Line 15" add passive microwave in front of "brightness temperature"

Section 1.5 What does it mean by: "The operational sea ice charts from the National Ice Center (NIC) are a relatively independent ..." this contradicts what you write in the text below these lines. See also in section 3.3 "The NIC ice charts are produced manually on the basis of satellite and reconnaissance data" Which satellite "pixel grid" are you referring to? Section 2.1 The title of this section is misleading. Microwave (and thermal) emission received by satellite sensors is attenuated by the atmosphere. In addition, emission from the atmosphere contributes to the radiation measured by the spaceborne sensors. Over the old sea ice and open water this contribution could be significant in comparison with the radiation from the sea surface. Line 26 What is "wind shear" ? Liquid water has much greater contribution to the measured passive microwave brightness temperatures than water vapour. For the sake of clarity: Section 2.2 should be placed before section 2.1 Section 2.2 Brightness temperatures for what type of ice are selected to be considered for a tie-point ? Section 2.3 How are the tie-point brightness temperatures are used to derive parameters in the Bootstrap and Bristol algorithms? Section 2.5 It is not clear what the authors call "uncertainty". Is it a quantity representing statistical variabilities or derived quantity deviation from an

accepted value? "ice concentration uncertainty of 1.4 % for the Bristol algorithm, and 1.7 % for the Bootstrap algorithm in frequency mode" compared to what? At what spatial and temporal resolutions?

Is this quantity relevant when discussing temporal and spatial variability and trends?

2.14 Level 4: Gap filling by spatial and temporal interpolation

This section contains interesting description of filling in missing data. The gap filling is required if the proposed data set to contain high temporal resolution sea ice information. If the goal is to have data sets for climatology analysis, is it necessary to go through these additional computations? In sections below, the authors discuss the hemispheric comparisons. Dis they actually filled temporal and spatial gaps in their data sets to carry out this analysis?

Section 3.1 Where did the sea ice climate data records came from? The authors claim that the discrepancy between the ice cover values they derived and NIC ice charts are due to the "atmospheric noise". They could have taken a look at historical weather or AVHHR/MODIS cloud imagery to confirm or deny that is the main cause for the differences. The increase in the discrepancy for the spring/summer seasons are very likely due to the algorithms do not accommodate seasonal variability in the emitting layer, e.g., melting snow, melt pond, change in the ice salinity. Statistics does not compensate for the physics of microwave radiation from different surfaces and propagation through the atmosphere.

Part of the section 3.1 is dedicated to discussion of differences between NIC ice charts and ESICR data sets. Why than there is a separate section "3.3 Ice chart and ESICR comparison discussion"? The last paragraph in section 3.3 should be placed into the conclusion and summary section

Section 3.4 Do the hemispheric sea ice cover and open trends described using ESIGR generated data sets differ significantly from those observed by using Bootstrap or

Bristol algorithms?

Please also note the supplement to this comment:
http://www.the-cryosphere-discuss.net/tc-2016-34/tc-2016-34-RC1-supplement.pdf

————————————————————

---

## Referee Comment (RC2) · W. Meier (Referee) · 22 Apr 2016

Summary

This manuscript documents the new EUMETSAT sea ice climate record, including the processing, quality control methods, and estimation of uncertainty. Results of the product are presented as trends in extent and, concentration, and days of open water. These results are consistent with similar analyses using other sea ice products, showing strong declines in the Arctic, and weaker increases in the Antarctic.

General Comment

The data product presented is a well-constructed product that will be beneficial to the community. Overall, this manuscript describes the product well and demonstrates its utility for monitoring climate change and variability. There are a few sections that could use some clarification and/or further details, which are noted below. The most signifi-

cant comment is that there are places where the text is quite repetitive – as if sections were written independently and then appended together; examples are given below. Also the writing style is somewhat unimaginative, making it read rather dull. The main example here is that nearly every figure is introduced in the text in paragraphs starting "Figure X shows..." or "Figure X is showing..." These read more like figure captions instead of a discussion of results. I would recommend rewriting many (if not all) of these to be more conversation. For example, instead of writing "Figure X shows Y has an increasing trend", simply write: "Y has an increasing trend (Figure X)" and then carry on the discussion from there.

One more issue is the section breakdowns. There are places where more nested subsections should be used. For example, Section 2.5 is "The sea ice concentration uncertainties", then Section 2.6 is "First component: instrument noise, algorithm and tie-point uncertainties". This is clearly a subsection under 2.5, so it shouldn't be listed at the same level, 2.6, but rather at something like 2.5.1 or 2.5a. I'm not sure of the specific style guidelines of The Cryosphere – maybe it doesn't allow more than one nesting of subsections? But even if so, I would then break out the contents until unnumbered sections under the main subsection, and denote them by underlines and/or italics.

Finally, some of the figures could be a bit more polished. Fore example, the legends in Figures 4-9 look more like working figures. Maybe do "Total Bias" instead of "total_bias". Also the x-axis labels are a bit messy – labels and ticks at regular intervals would be better – e.g., at the beginning of a month, not varying days within months; particularly for Figures 4-7, where the date labels seem rather random. And the y-axis labels in Figure 8 should be outside the axis, not inside Also, remove the straight lines across missing data is Figures 6 and 7. There is also inconsistent marker styles (line vs. no-line) in Figures 8 and 9. Finally, the fonts for Figures 4-9 could be a little nicer – they low almost dot-matrix printed.

Much of these suggestions are style rather than science, but I still think making the

suggested changes will help the manuscript convey the science better. These shouldn't be too difficult to accomplish; my decision is Minor Revision.

Specific Comments (by page/line number):

1/30: There are several other, more recent papers on the increasing Antarctic sea ice, e.g. Goose and Zunz, The Cryosphere, 2014; Holland, GRL, 2014; Li and Holland, Nature, 2014; Holland and Kwok, Nature Geoscience, 2012.

2/1-2: It says the extent is defined by concentration >15% as in Parkinson and Cavalieri (2008), but later a 30% threshold is used. The text should be made consistent.

3/9: What are "Dicke" radiometers? Not sure this detail is necessary.

3/14: Explain why SMMR data are produced only every other day.

3/23: RSS is now distributing Version 7. Understand that Version 6 was used here, but would be good to mention the new version. And will Version 7 eventually be used (e.g., a future reprocessing)?

4/4: Where is the SSMIS L2B NRT data stream obtained?

4/22: Units for "220" and "60"?

4/23-24: The approach of using 100% and 0% tiepoints for brightness is not uncommon for visible imagery. However, there is potential error in that new ice growth will be darker, even at 100% concentration. So the tiepoint approach will underestimate new ice. This is a relatively small area (especially during spring-fall when there is visible light), so I don't think it affects the analysis significantly, but would be worth mentioning.

4/29: I believe the official designation should be "U.S. National Ice Center"

5/1: Where were the charts obtained – directly from NIC? NSIDC also has a database of gridded charts (http://nsidc.org/data/g02172), so it's good to be precise on the data citation. And while these are operational charts, there are a couple (older) peer-

TCD

reviewed journal articles that could be cited: Dedrick et al., Canadian J. Remote Sensing, 2001; Partington et al., JGR, 2003.

5/1-20: This is an area where the text is repetitive and could be reorganized a bit to be more concise. For example, on line 1, it says the charts are produced on "a regular basis", then on line 17, it says "a weekly compilation" – why not just say they're produced on a "weekly basis" on line 1 and eliminate the later sentence. Similarly, lines 7-8 say the source is "primarily satellite imagery", then in lines 11-17 the specific satellites are used and the caveat that ship and aircraft observations are also used.

5/20: Missing, or at least not clear, is that the chart concentrations are given in ranges of 10ths. This is particularly important because the PM concentrations are to the nearest %. It's not really explained how this is handled and how it affects the comparisons. For NIC values, is the midpoint used as the concentration (e.g., for 7-8 tenths, is a value of 75% used for comparison), or are the PM values binned into the NIC increments?

8/10-28: Another place with the section nesting doesn't seem quite right. Maybe just remove the 2.4 designation and merge with 2.3, which is very short. There is also a lot of repetition of text in the two sections – for example, line 5 is exactly repeated in line 28. Also, lines 11-14 (and page 9, lines 1-2) repeat much of the text in 2.3.

9/28 – 10/1-3: repeats 9/21-22.

10/6: Section 2.7 is really part of the error in 2.6, right (i.e., the "First component")? Then it shouldn't really be a separate section at the same level as 2.6 (and the following "Second component").

10/26: Again, isn't smearing (2.9) part of the representativeness error (2.8)?

12/14: What are the monthly varying ocean masks? Where are they from, how are they used? This is later explained in 2.13 (I think) but the explanation should be put with or before its first use here.

13/29-30: the land-spillover correction by Cavalieri et al. (1999) is a bit more involved

than described and includes not just grid cells along the coast (i.e., directly adjacent to a land grid cell), but also grid cells one and two pixels away from the coast, with different corrections applied to each. It's okay to leave the details to the reference, but as it's written now, the description is somewhat misleading.

14/11: I think you mean "Grid cells with missing data are filled with interpolated values only in the level 4 processing..." As it reads, it sounds as if one might consider interpolating and filling cells with data.

14/12: "Daily coverage is never complete..." – is this due to the pole hole? Even so it doesn't really go with the rest of the sentence "...and occasionally there are missing scan lines..." Maybe "Daily data coverage is never complete due to the hole near the North Pole and occasionally there are missing scan lines...."

14/16 – 15/12: This interpolation procedure description seems unnecessarily complex and difficult to follow, and some parameters in the equation are not defined (e.g., what are "W" and "w"). It's okay to show the specific equations, but there should also be a clear text explanation the approach (e.g., similar to what is done for the blending of Bootstrap and Bristol concentration – the equations are given, but it also simply says a weighted average with equal weighting at 40% concentration).

15/20: Suggest removing "It is clear".

15/26: repeat of 15/16

15/24: Here is where the issue of the NIC use of 10ths bins is salient – how is this handled? Also, the charting changed quite a bit when Radarsat-1 started to be used in 1995 – the charts got less dependent on passive microwave, more detailed, and probably more accurate. So I think pre-RS1 and post-RS1 might have quite different characteristics in comparisons with the PM data. Was this looked at? I don't see anything obvious in Figure 4, but I would expect some difference.

16/8-9: Ocean cells have above 0 concentrations even with the atmospheric corrections? What about the smearing error – couldn't that contribute by smearing out the ice into open water areas near the ice edge?

16/30: National Ice Center should be capitalized.

17/1: The missing chart period is noted for (at least) the third time. It's okay to reiterate the point for clarity, but each time it's written as if for the first time. At least rewrite to "As noted earlier, there were no digital ice charts available..." But I don't think it's needed at all here.

17/18: The overlap is during July and August, which isn't during the minimum. And I don't think the limited number of ice data points is the main factor here – it's that July and August is a time of significant melt and melt pond coverage.

17/30: Repetitive of earlier description of the NIC charts – don't need to say it again here.

18/2: Who are the "two other producers"? Is this referring to other ice charting centers?

18/10-11: "...yield the same concentrations..."

18/30: Why is the 30% concentration used to define extent? It's perfectly reasonable, but why use that instead of the more common 15% threshold?

19/16: The 5% and 95% ranges need to be explained. I would assume they describe the range between the highest and lowest 5% values. But then how in Figure 10 is the 2012 extent within the 5% contour when it is the lowest extent in the record?

20/1: Recommend citing Parkinson, GRL, 2014, who calculated open water season trends for the Arctic from passive microwave sea ice fields.

20/7-10: This sentence is a bit unclear. I think I understand – different periods were tested and 5 days was the shortest period found that gave reasonably stable values without much noise. Right?

21/23-29: This seems odd to me. In Figure 14, the open water trend near the edge is +30 days for 2004-2014, but there have been record high extents during the same period? I guess I can see larger extents for shorter periods, but the large number seems unlikely – the open water season is over a month longer, but still ice reaches record maximums? Maybe this mostly points to the shortness of the period. Maybe it's just one year (or two) that are really anomalous that dominate the signal?

---

## Referee Comment (RC3) · G. Heygster (Referee) · 29 Apr 2016

The manuscript presents a harmonized Arctic and Antarctic sea ice concentration data set based on passive microwave SMMR, SSM/I and SSMIS satellite observations from 1978 to 2014, spanning observations from 11 different instruments. The new things of this data set are the atmospheric correction and uncertainties given for each pixel individually, based on a careful error estimation. The various error components are considered separately and an overall error model is forwarded in Fig. 3.

General comments:

It is an elegant approach to use dynamic tie points to compensate for sensor drift and inter-sensor calibration. Were time series of the tie points also investigated? Could a seasonal cycle, drift or jumps with sensor change be observed? P(age)4L(ine)4 SSMIS data: While the SSM/I data from RSS comprise corrections for geo-location,

sensor calibration and inter-sensor calibration, this is not the case for the SSMIS data. Were similar corrections applied to the SSMIS data? Is the dynamic tie points method considered to compensate automatically these errors?

The large number of 14 subsections of Section 2 could be more structured for easy understanding. I would consider Secs. 2.6 to 2.10 as subsections of 2.5 Uncertainties. According to the short Sec. 2.5 and Eq. (1) I would expect two subsections, one on the tie point uncertainty and one on the representativeness error, with perhaps more subsections. From the sentence in Sec 2.5 'The tie point uncertainty $\varepsilon\_$tie-point , including residual atmospheric noise, sensor noise and ice surface emissivity variability, is derived from measurements as the first component of uncertainty' I would expect the section on the first component to treat exactly these errors in exactly this order. However, the heading of 2.6 is 'First component: instrument noise, algorithm and tie point uncertainties', which is different. If the error components are not treated in the mentioned order, then the relation between the different error contributions should be made clear. I do not find algorithm noise treated in Sec. 2.6 which is promised in the heading. Rather, is seems to be treated in Sec. 2.10. Moreover, Secs. 2.12 to 2.14 deal with the correction of known errors which could be a common heading for these three.

Sec. 1.5 Ice charts:

It is very good to use ice chart as independent source. The data source should be indicated. Are those data freely available? Reference? Are they provided in the same polar stereographic grid as the SSM/I data? Is there any conversion required, e.g. from region shapes to pixel data? Some of the ice chart errors can be quantified: If the charts give ice concentrations in 10% steps, this corresponds to adding quantization noise equally distributed in the IC interval [0,10]. It has a standard deviation of 10*sqrt(1/12) = 2.9% IC. Moreover, weekly ice charts will contain as error the development of the ice within one week. This can be estimated considering ice chart differences of successive weeks.

Sec. 2.7 Geo-location error: A reference to Hollinger et al. (1990) is not suitable for SSMIS geolocation which was not launched at the time of publication. More suitable would e.g. be Poe et al., Geolocation Error Analysis of the Special Sensor Microwave Imager/Sounder, IEEE TGRS, VOL. 46, NO. 4, APRIL 2008. They find a geolocation error 'in excess of 20–30 km' which cannot be considered small compared to the footprint size. Was the SSMIS L2B NRT data version you are using (P4L4) ever corrected for such errors?

It would be helpful for comparison with other publications (e.g. with Spreen et al. 2008: . J. Geophys. Res 113, C02S03, doi:10.1029/2005JC003384) to give, in addition to the time series of bias and stddev in Figs. 4-7, the average values for these quantities.

It could be interesting to show in Figs. 8 and 9 also the open water bias for the overlap period. Data Levels 3 and 4 at the beginning of Sec. 2.11 should be briefly explained.

As TCD does no more any typesetting, it would be nice if the manuscript would obey simple rules of typesetting: numbered quantities are denoted with capital words like Figure 1, Table 3, etc., and numbers up to ten are written as words.

Minor points:

Use for the same quantity always the same symbol. Wind is sometime u*, sometimes U* . Ice concentration sometime ic, sometimes IC. Does the * symbol in Eq. (1) at u* have a meaning?

P3L14 'There is SMMR data only every second day': No, each day. Only, because of the narrow swath, full coverage requires data accumulation over two days. Similar P15 L11.

P6L13 ' The representation of atmospheric liquid water column in the NWP data is not suitable to use for brightness temperature correction.': Give reason: because of the spatial and temporal variability of cloud, which is higher than the model grid cell size and model time step size.

[Figure]

P7L1 '. The fluctuations due to atmospheric and surface emission are systematic.': Meaning unclear.

P8 L7: explain OSI SAF

P9L23 'spatial ice concentration standard deviation': meaning of 'spatial' unclear. Omit?

P11L14: meaning of 'logically' unclear. Omit?

P11L30 to P12L3: Necesscity of Eq.(6), Boxcar and Heaviside function unclear. For the subsequent text, it is sufficient to define the truncated ice concentration alpha by P12L4-6.

P13 L15 weighting function Eq. (12): In principle, the function should reflect the antenna pattern. Then it would have to be a Gaussian. Linear weights taken here as simple approximation.

P14L22 and P15L2: equation 6 -> Eq. (13)

P15L18 Latter -> later

P19L22 '(Q3)' not needed, never used.

P20L1-5: There is an extended literature on detecting onset and end of melt season with sophisticated methods. It is ok to use a simple method here, but it should be mentioned that more exist, e.g. by citing the recent article by Close et al.: Regional dependence in the timing of onset of rapid decline in Arctic sea ice DOI: 10.1002/2015JC011187. See also literature cited there.

P21L5: trend in number of open water days shown in figure 13 → Figure 12.

P21 L13 . As for the Arctic the open water days is calculated.. → As for the Arctic, the open water days are calculated.

P21 L 29 the pole → Antarctica or the South Pole

P23 L16 'The next update': insert 'of the ESICR data set'

P23 L9 'minor differences': why should there be any differences between the two algorithms? Specify.

---

## Author Comment (AC1) · 17 Jun 2016

Thanks for the thorough review and for helping us making this MS a better paper. Please find our reply to all of your comments below.

General comments: The content of this publication can be considered very important contribution to the creation of consistent several decades span of sea ice cover data sets. This article reads like it was written by several different people. Up to the section 2.1 the writing style and terminology used do not explain clearly the data sets properties used in this project. The list of recommendations below will improve the readability of these sections. Thanks for providing these recommendations. We will modify the text accordingly.

Starting with section 2.1 the writing style and the clarity greatly improves and very few minor clarifications are needed. But, very important section 3.3 Ice chart and ESICR

comparison discussion is not easy to read. The comparison should be better organized. The authors should spend some time on deciding what will be the primary use of these long term data sets. The spatial and temporal resolutions, as well as the accuracy of retrievals are usually driven by these applications. The criteria for comparisons with existing data sets and/or ice charts are also dictated by who will be using this new and improved sea ice cover information. Making trend analysis using different sea ice concentration algorithms shows quite different negative trends of the Arctic sea ice cover. This indicates that the geophysical noise sources (including atmospheric water vapor ice types etc.) have climatic trends as well as the ice extent and that the uncertainties are significant and should be quantified. This was the motivation for doing this reprocessing and it has been the aim to create as long a record as possible thus using 19 and 37 GHz channels with the spatial resolution that these channels have.

Algorithms designed for processing spaceborne measurements are developed for specific applications. Some of these algorithms are very simple and set up to provide yes/no indicators to mask ice covered (>threshold value) areas. Some other algorithms, especially to be used in near-real time for navigational support, are more complex and are region and season specific. The regional and seasonal variability of sea ice cover can be partially compensated by using seasonal, regional algorithm parameters and sea ice signatures. These parameters can be derived using RTM. The complexity of the microwave radiation from the sea ice cover must be accommodated. The authors do state that understanding what contributes to the emission from the ocean surface (e.g., snow free ice floes, ice flows covered heavy wet snow, young saline ice, etc..etc„) is very important but do not carry out their evaluations region by region. The "new" hybrid algorithm should have region specific "flagging" to allow global processing. Both the noise reduction using NWP data and and an RTM on every measurement and the tie-points are region-specific so that the distribution of ice types (meaning sea ice with different emissive characteristics) is accounted for regionally. The seasonal variability including melting snow on sea ice is to some extent accounted for using the dynamical tie-points.

The authors should dedicate more discussion time in planning their future work to regional seasonal variability of the ice cover and how that impacts microwave and optical brightness temperatures. We are already in the process of planning the next updates of the dataset and we have already included these plans in the text in 4.1.

The authors have selected a hybrid algorithms that uses fixed coefficients multiplied by observed brightness temperatures. It is not clear how using NWP is incorporated into calculations of these coefficients or in computations of dynamic tie-points. The text has been revised with attention to better explain the use of tie-points and atmospheric correction. The Bristol algorithm is using fixed coefficients only to rotate the coordinates of the plane which is spanned in the space of the three brightness temperatures (19v, 37v, 37h) that the algorithm is using. The coefficients used for quantifying the sea ice concentration are the tie-points (or ice line coefficients) and these are dynamic. The bootstrap algorithm is not using fixed coefficients for rotation only the dynamical tie-points. These two algorithms are used in combination.

All throughout descriptions of different algorithms the authors discuss items they call "uncertainties". It is not clear what this particular quantity represents. Is it a statistical value or derived value related to variability (or accuracy) in measured brightness temperatures or other measured (or derived) parameters? In a real world there are very few instruments that can claim to have accuracy of less than 2%. The sea ice concentration uncertainty is quantifying the sea ice concentration uncertainty on every pixel as an absolute quantity and given in percent. It describes the independent instrument noise, the geophysical random noise and the representativeness error. The total uncertainty is normally higher than 2%.

The sea ice cover data sets generated using MODIS and/or SAR provide higher spatial resolution data but the authors should not forget that the algorithms used to generated ice cover imagery may contain sensor/processing algorithm specific errors, e.g., ocean surface roughness effects on SAR imagery; cloud cover on MODIS The authors have acknowledged the need for the algorithms to accommodate pack ice and marginal ice

regions. It is not clear if the hybrid algorithm will be tested for these diverse ice cover regions. It is also not clear how different coefficients used in this new algorithm can be derived using physical properties of different types of ice and weather conditions.The ice cover in marginal ice zones can change very quickly depending on the wind (e.g., read about icebreakers trapped in Barents ice; ships stuck in Weddell sea) and in some parts of the world the extent of the ice cover is dictated by the thermal currents (e.g., sea ice cover in Labrador and Newfoundland waters). The data used for generating the ice charts such as microwave radiometers, scatterometers, visual/infrared scanners, SAR instruments etc. each have uncertainties and ambiguities. Combining different data in the analysis may to some extent reduce the ambiguities, but the manual interpretation introduces other uncertainties. A discussion of the ice chart uncertainties is included in section 3.1 and the text has been revised to state this more clearly. Ice charts include these different regions mentioned above and we have included a discussion of the uncertainties in ice charts (sections 1.5 and 3.1).

The authors have discussed "smearing" of ice cover location but did not address "smearing" effects of orbital data gridding onto a fixed grid. These effects can be determined by comparing retrievals from ascending and descending satellite overpasses looking at ice edge that was not changed (in 10 to 12 hours between satellite passes) by prevailing winds We have developed a model for the "smearing" for this dataset using an radiometer imaging simulator. The smearing uncertainty is one of the two components in the total sea ice concentration uncertainty. The gridding uncertainty when binning swath data onto a predefined grid for example quantified by the daily grid point STDDEV is not independent from the tie-point uncertainty component and we have not been able to include it here.

The authors should have more systematic plan for testing their chosen algorithm. Signature areas of predominant sea ice type cover (e.g., Sea of Okhotsk for seasonal ice cover; old ice covered areas; mid-winter Labrador Sea ice cover) are ideal for testing algorithms. The algorithm testing is described in separate studies e.g. Andersen et

al. (2006), Andersen et al. (2007) and in Ivanova et al. (2015). These references are given in the text. . . Section specific comments: In the abstract 1. Replace: American by NASA, NOAA, or NSIDC depending which agency was responsible for the satellite sensor, satellite launch, and satellite data processing Sea ice area by sea ice covered area OK.

2. Sentence starting on line 16: "The methodology. . . . . . . .and ending on line 21 is not clear It has been reformulated.

3. Are the authors saying that in preparation of this sea ice dataset : 1) NWP and RTM were used to reduce impact of the atmospheric conditions on the measured brightness temperatures; 2)to reduce the inter-sensor comparison bias dynamic tie points were used in retrieval algorithms, and 3) hybrid algorithm was created using Bristol and Bootstrap algorithms. Yes, that is true. We have tried to clarify this in the text.

This algorithm was used in estimation of the spatial and temporal variabilities in sea ice concentration retrieval accuracy. The algorithm was used for generating the sea ice concentration dataset.

4. Who generated "sea ice charts from the Arctic and the Antarctic" (lines 22 and 23) We have added: "U.S. National Ice Center" as the source. References to the datasets that we have used have been included in the text.

5. What are the "intermediate ice concentration" We have specified that now [between 1 and 99%].

6. Line 32: see comment #1 OK. We have specified the agencies.

7. Pg. 2, line 1: by "Here the sea. . ." The authors are saying In publications referenced above the sea ice. . ... OK. We have replaced "Here" with "Throughout this paper"

8. Pg. 2 line 8: replace "atmospheric parameters" by weather conditions such as wind generated ocean surface roughness and cloud cover OK. we have done that throughout the text.

9. What is "structural uncertainties" We have deleted the term because it is not really needed. Structural uncertainty is different outcome from different methods using the same data.

10. Line 15 Define "noise" OK, good point. Noise is the Tb fluctuations caused by the electronic components in the instrument, ice and water surface emissivity, and weather conditions, or anything which is not the sea ice concentration variability.

11. Why "climatic trends" are "artificial trends" ??? They are artificial trends because they appear as trends in the sea ice concentration but they are due to something else. This has been clarified in the text.

12. Be more specific in "for the properties that we are able to quantify,????? OK. "NWP data: wind, air temperature and atmospheric water vapour" it has been included in the text.

13. Please explain what are the "dynamical tie-points" OK. It has now been explained in the introduction.

14. "residual uncertainties" for which parameters and please, define uncertainty is it - accuracy. It is sea ice concentration accuracy. This has been specified in the text.

In section 1.1 Line 14 after "84°." Should be replaced by: "SMMR data were acquired every second day" OK. "...because of satellite power limitations" has been added.

In Section 1.2 "The SSMIS data (used in this project?) are from the L2B near real time data-stream" received from??? Received via EUMETCast, initially processed at NOAA. This has been specified in the text.

In section 1.3: Please clarify your statement on lines 7 and 8. Was sea ice coverage included in NWP? If yes, where did that information come from? The sea ice coverage is included in the ECMWF numerical weather prediction model. These data are from the Hadley Centre: a compilation of ice charts from different ice centers and microwave radiometer data processed by J. Comiso, NASA.

In section 1.4 line 15 insert "The coarse resolution of the passive microwave brightness. . ." OK Line 22 and line 24 "Brightness above 220 (units?) 60 (units?) The pixels have brightness values between 0 and 255. These values are not calibrated to a physical quantity for brightness. Here they are used in a relative sense to identify pixels with sea ice and open water. We have included a short discussion of the albedo variability of water and ice. It is short because this is beyond the point here. The purpose of the image is only to provide realistic input data to the imaging simulator in terms of floe size distribution, the distribution of cracks and leads etc. The MODIS image concentration is only used as input to the simulator and not compared to coincident microwave radiometer data or other sea ice concentration estimates.

"We use 6 hourly data at a resolution of 1.25 degrees." Why this temporal and spatial resolution was used? This is the resolution of the model. The NWP data were interpolated to the satellite swath data.

Section 1.4 Line 15" add passive microwave in front of "brightness temperature" OK

Section 1.5 What does it mean by: "The operational sea ice charts from the National Ice Center (NIC) are a relatively independent . . ." this contradicts what you write in the text below these lines. OK, we have rewritten this section and taken your comment into account.

See also in section 3.3 "The NIC ice charts are produced manually on the basis of satellite and reconnaissance data"

Which satellite "pixel grid" are you referring to? This the the EASE grid where the sea ice concentration data is provided. This has been specified.

Section 2.1 The title of this section is misleading. Microwave (and thermal) emission received by satellite sensors is attenuated by the atmosphere. In addition, emission from the atmosphere contributes to the radiation measured by the spaceborne sensors. Over the old sea ice and open water this contribution could be significant in

comparison with the radiation from the sea surface. It is very true that the surface emissivity variability does contribute to the sea ice concentration uncertainty. However, the particular processing step which is described in section 2.1 is only reducing the uncertainty due to the weather conditions which are quantified by the NWP model.

Line 26 What is "wind shear" ? We have deleted "shear".

Liquid water has much greater contribution to the measured passive microwave brightness temperatures than water vapour. The cloud liquid water (CLW) is for sure one of the important error sources. However, it is difficult to correct for using NWP data because its spatial and temporal variability is not quantified very well by the model. We have selected a sea ice concentration algorithm over open water which has low sensitivity to CLW.

For the sake of clarity: Section 2.2 should be placed before section 2.1 Section 2.2 OK.

Brightness temperatures for what type of ice are selected to be considered for a tie-point ? Tie-points are selected for pixels where the NASA team ice concentration is greater than 95% and south of 84N in the Arctic in order to be consistent with the SMMR coverage. This includes both multiyear ice and first-year ice, but not new ice.

Section 2.3 How are the tie-point brightness temperatures are used to derive parameters in the Bootstrap and Bristol algorithms? The tie-points are reference points for 0% and 100% ice. This has been clarified in the text.

Section 2.5 It is not clear what the authors call "uncertainty". It has been defined in the beginning of section 2.5. Whenever referring to uncertainty it is the sea ice concentration uncertainty.

Is it a quantity representing statistical variabilities or derived quantity deviation from an accepted value? Yes, the sea ice concentration uncertainty is computed using our model for uncertainty which is described in section 2.5.

"ice concentration uncertainty of 1.4 % for the Bristol algorithm, and 1.7 % for the Bootstrap algorithm in frequency mode" compared to what? At what spatial and temporal resolutions? The instrument noise is an absolute error given in the same units as the ice concentration [percent]. The temporal resolution is given by the integration time of the instrument. Is this quantity relevant when discussing temporal and spatial variability and trends? The instrument noise is an independent error included in the tie-point uncertainty component (one of the two uncertainty components). The magnitude of this uncertainty is nice to know because it sets a lower level for the sea ice concentration uncertainty.

2.14 Level 4: Gap filling by spatial and temporal interpolation This section contains interesting description of filling in missing data. The gap filling is required if the proposed data set to contain high temporal resolution sea ice information. If the goal is to have data sets for climatology analysis, is it necessary to go through these additional computations? The step from level 3 (where no interpolation is done) to level 4 with temporal and spatial interpolation is not necessary but it increases the number of users of the data. The interpolated values are flagged so that it is possible to identify these as originating from interpolation.

In sections below, the authors discuss the hemispheric comparisons. Dis they actually filled temporal and spatial gaps in their data sets to carry out this analysis? Yes, the analysis is done on L4 data to avoid latitude-biases from coverage frequency (mostly relevant for SMMR).

Section 3.1 Where did the sea ice climate data records came from? The authors claim that the discrepancy between the ice cover values they derived and NIC ice charts are due to the "atmospheric noise". They could have taken a look at historical weather or AVHHR/MODIS cloud imagery to confirm or deny that is the main cause for the differences. This is "atmospheric noise" which is causing the discrepancies. AVHRR/MODIS data does not quantify water vapor, cloud liquid water, wind speed which are the major error sources over open water together with surface temperature. Sea surface temperature can of course be quantified using the infrared channels on

these two sensors but it is not the primary error source.

The increase in the discrepancy for the spring/summer seasons are very likely due to the algorithms do not accommodate seasonal variability in the emitting layer, e.g., melting snow, melt pond, change in the ice salinity. Statistics does not compensate for the physics of microwave radiation from different surfaces and propagation through the atmosphere. The dynamical tie-points that we use do compensate for seasonal and interannual changes in the sea ice brightness temperature signature globally. The local and regional variability in the signatures can, however, not be accounted for and does increase the sea ice concentration uncertainty. However, this increased uncertainty level during melt is quantified in the uncertainty estimate given for every pixel.

Part of the section 3.1 is dedicated to discussion of differences between NIC ice charts and ESICR data sets. Why than there is a separate section "3.3 Ice chart and ESICR comparison discussion"? The last paragraph in section 3.3 should be placed into the conclusion and summary section

Section 3.4 Do the hemispheric sea ice cover and open trends described using ESIGR generated data sets differ significantly from those observed by using Bootstrap or Bristol algorithms? We have moved the end of 3.3 to the conclusions according to your suggestion. It is clear that the Bootstrap, the Bristol and the ESICR is producing different trends given the same data. In the ESICR dataset the Bristol is used over ice (because it performed best over ice among all algorithms (Andersen et al. 2007)) and the Bootstrap is used over open water (because it performed best over water among all algorithms (Andersen et al. 2006)). This combination minimizes the sensitivity to noise overall.

Please also note the supplement to this comment:
http://www.the-cryosphere-discuss.net/tc-2016-34/tc-2016-34-AC1-supplement.pdf

---

## Author Comment (AC2) · 17 Jun 2016

Thanks for the constructive comments to our MS and helping us making this a better paper. We have tried to accommodate nearly all. We have reprocessed the ESICR and ice chart comparison using the published NIC dataset at NSIDC and NIC as you have suggested. See specific answers in the text below.
This manuscript documents the new EUMETSAT sea ice climate record, including the processing, quality control methods, and estimation of uncertainty. Results of the product are presented as trends in extent and, concentration, and days of open water. These results are consistent with similar analyses using other sea ice products, show-

[Figure]

ing strong declines in the Arctic, and weaker increases in the Antarctic.

General Comment The data product presented is a well-constructed product that will be beneficial to the community. Overall, this manuscript describes the product well and demonstrates its utility for monitoring climate change and variability. There are a few sections that could use some clarification and/or further details, which are noted below. The most significant comment is that there are places where the text is quite repetitive – as if sections were written independently and then appended together; examples are given below.

Also the writing style is somewhat unimaginative, making it read rather dull. The main example here is that nearly every figure is introduced in the text in paragraphs starting "Figure X shows..." or "Figure X is showing..." These read more like figure captions instead of a discussion of results. I would recommend rewriting many (if not all) of these to be more conversation. For example, instead of writing "Figure X shows Y has an increasing trend", simply write: "Y has an increasing trend (Figure X)" and then carry on the discussion from there. We have tried to follow you guidelines on style.

One more issue is the section breakdowns. There are places where more nested subsections should be used. For example, Section 2.5 is "The sea ice concentration uncertainties", then Section 2.6 is "First component: instrument noise, algorithm and tie-point uncertainties". This is clearly a subsection under 2.5, so it shouldn't be listed at the same level, 2.6, but rather at something like 2.5.1 or 2.5a. I'm not sure of the specific style guidelines of The Cryosphere – maybe it doesn't allow more than one nesting of subsections? But even if so, I would then break out the contents until unnumbered sections under the main subsection, and denote them by underlines and/or italics. We have restructured section 2 with subsections as you suggest.

Finally, some of the figures could be a bit more polished. For example, the legends in Figures 4-9 look more like working figures. Maybe do "Total Bias" instead of "total_bias". Also the x-axis labels are a bit messy – labels and ticks at regular intervals

would be better – e.g., at the beginning of a month, not varying days within months; particularly for Figures 4-7, where the date labels seem rather random. And the y-axis labels in Figure 8 should be outside the axis, not inside Also, remove the straight lines across missing data is Figures 6 and 7. There is also inconsistent marker styles (line vs. no-line) in Figures 8 and 9. Finally, the fonts for Figures 4-9 could be a little nicer – they low almost dot-matrix printed. You are right! These figures have been replotted and the ice chart comparison reprocessed using publicly available and documented data sets. The period has been extended to cover the entire dataset period (except the ice chart data gab on the southern hemisphere 1994-2006).

We have left out the "total" bias and std from the plots. The quantity was dependent on the number of ice and water points and so not so meaningful.

Much of these suggestions are style rather than science, but I still think making the suggested changes will help the manuscript convey the science better. These shouldn't be too difficult to accomplish; my decision is Minor Revision.

Specific Comments (by page/line number): 1/30: There are several other, more recent papers on the increasing Antarctic sea ice, e.g. Goose and Zunz, The Cryosphere, 2014; Holland, GRL, 2014; Li and Holland, Nature, 2014; Holland and Kwok, Nature Geoscience, 2012. Thanks. We have updated the references.

2/1-2: It says the extent is defined by concentration >15% as in Parkinson and Cavalieri (2008), but later a 30% threshold is used. The text should be made consistent. We agree this is inconsistent! The threshold used in this paper is 30%. This threshold was selected for the example applications of the dataset in section 3.4 because it gave the most stable and least noisy results. That the threshold is higher than in Parkinson and Cavalieri (2008) maybe because we are not using weather-filters in the processing and occasionally there is open water noise over 15% (as shown in the ice chart comparison).

3/9: What are "Dicke" radiometers? Not sure this detail is necessary. Ok, it is deleted.

3/14: Explain why SMMR data are produced only every other day. Ok, it is included in the text.

3/23: RSS is now distributing Version 7. Understand that Version 6 was used here, but would be good to mention the new version. And will Version 7 eventually be used (e.g., a future reprocessing)? Ok. It has been included in the text. The next reprocessing will be done using the EUMETSAT climate SAF dataset, even though there are still some open issues about the geolocation correction to be resolved.

4/4: Where is the SSMIS L2B NRT data stream obtained? The SSMIS L2B NRT data are obtained from EUMETSAT via the EUMETCast distribution system, in BUFR format. This distribution by EUMETSAT is a redistribution of data from NOAA.

4/22: Units for "220" and "60"? The brightness in this image gallery is non-calibrated number between 0 and 255 and it does not have unit. This has been explained in the text.

4/23-24: The approach of using 100% and 0% tiepoints for brightness is not uncommon for visible imagery. However, there is potential error in that new ice growth will be darker, even at 100% concentration. So the tiepoint approach will underestimate new ice. This is a relatively small area (especially during spring-fall when there is visible light), so I don't think it affects the analysis significantly, but would be worth mentioning. Ok, it has been included in the discussion. The intention with the classification is to create a realistic spatial distribution of sea ice for input to the imaging simulator. We selected an image covering a large area with both 100% ice and open water and all concentrations in between to compare the actual ice concentration as given in the image with the simulated ice concentration using the imaging simulator and the spatial resolution of the microwave radiometer. The mismatch between these two is the smearing. We found that the smearing was relatively uniform in between the 0% and 100%. Logically the smearing is zero at the tie-points.

4/29: I believe the official designation should be "U.S. National Ice Center" Ok, it has

been changed.

5/1: Where were the charts obtained – directly from NIC? NSIDC also has a database of gridded charts (http://nsidc.org/data/g02172), so it's good to be precise on the data citation. And while these are operational charts, there are a couple (older) peer reviewed journal articles that could be cited: Dedrick et al., Canadian J. Remote Sensing, 2001; Partington et al., JGR, 2003. The reference has been included. We have reprocessed the ice chart intercomparison with the NSIDC and NIC datasets and extended the comparison to cover 1978-2015. The datasets that we have used have been specified in the text.

5/1-20: This is an area where the text is repetitive and could be reorganized a bit to be more concise. For example, on line 1, it says the charts are produced on "a regular basis", then on line 17, it says "a weekly compilation" – why not just say they're produced on a "weekly basis" on line 1 and eliminate the later sentence. Ok, we have done as suggested.

Similarly, lines 7-8 say the source is "primarily satellite imagery", then in lines 11-17 the specific satellites are used and the caveat that ship and aircraft observations are also used. Ok, we have deleted the redundant sentences.

5/20: Missing, or at least not clear, is that the chart concentrations are given in ranges of 10ths. This is particularly important because the PM concentrations are to the nearest %. It's not really explained how this is handled and how it affects the comparisons. For NIC values, is the midpoint used as the concentration (e.g., for 7-8 tenths, is a value of 75% used for comparison), or are the PM values binned into the NIC increments? OK, we have redone the intercomparison using the data that are publicly available and documented on NSIDC and NIC. This has also resulted in a longer validation until 2015. We have included references to these documents data and procedures.

8/10-28: Another place with the section nesting doesn't seem quite right. Maybe just remove the 2.4 designation and merge with 2.3, which is very short. There is also a

lot of repetition of text in the two sections – for example, line 5 is exactly repeated in line 28. Also, lines 11-14 (and page 9, lines 1-2) repeat much of the text in 2.3. Ok, we have merged the sections and deleted the redundant sentences according to your suggestions.

9/28 – 10/1-3: repeats 9/21-22. Ok, 9/21-22 has been deleted.

10/6: Section 2.7 is really part of the error in 2.6, right (i.e., the "First component")? Then it shouldn't really be a separate section at the same level as 2.6 (and the following "Second component"). Ok, the "geo-location error section" is now part of the "first component . . ."

10/26: Again, isn't smearing (2.9) part of the representativeness error (2.8)? Yes, 2.9 has been made a subsection.

12/14: What are the monthly varying ocean masks? Where are they from, how are they used? This is later explained in 2.13 (I think) but the explanation should be put with or before its first use here. The monthly varying ocean masks are based on the NSIDC monthly maximum ice extent climatology, available at http://nsidc.org/data/smmr_ssmi_ancillary/ocean_masks.html. The extent in these ocean masks have been extended with 100 km (in all directions). For the extension of the data record from 15.10.2009 the extension distance was increased from 100 km to 300 km for the Southern Hemisphere, as the ice extent has increased in some areas in the recent years.

13/29-30: the land-spillover correction by Cavalieri et al. (1999) is a bit more involved than described and includes not just grid cells along the coast (i.e., directly adjacent to a land grid cell), but also grid cells one and two pixels away from the coast, with different corrections applied to each. It's okay to leave the details to the reference, but as it's written now, the description is somewhat misleading. Ok, we have revised this section.

14/11: I think you mean "Grid cells with missing data are filled with interpolated values only in the level 4 processing..." As it reads, it sounds as if one might consider interpolating and filling cells with data. Ok, the "only" has been deleted.

14/12: "Daily coverage is never complete..." – is this due to the pole hole? Even so it doesn't really go with the rest of the sentence "...and occasionally there are missing scan lines..." Maybe "Daily data coverage is never complete due to the hole near the North Pole and occasionally there are missing scan lines...." Ok, it has been changed according to your suggestion.

14/16 – 15/12: This interpolation procedure description seems unnecessarily complex and difficult to follow, and some parameters in the equation are not defined (e.g., what are "W" and "w"). It's okay to show the specific equations, but there should also be a clear text explanation the approach (e.g., similar to what is done for the blending of Bootstrap and Bristol concentration – the equations are given, but it also simply says a weighted average with equal weighting at 40% concentration). Ok, we have restructured this section with clear explanations of the procedures.

15/20: Suggest removing "It is clear". Ok

15/26: repeat of 15/16 Ok, 15/26 is deleted.

15/24: Here is where the issue of the NIC use of 10ths bins is salient – how is this handled? Also, the charting changed quite a bit when Radarsat-1 started to be used in 1995 – the charts got less dependent on passive microwave, more detailed, and probably more accurate. So I think pre-RS1 and post-RS1 might have quite different characteristics in comparisons with the PM data. Was this looked at? I don't see anything obvious in Figure 4, but I would expect some difference. We haven't done an in depth analysis of this. The open water bias does seem to have a decreasing trend. We think that this might be due to better quality NWP data from 1978 to 2015. It has been discussed in the text.

16/8-9: Ocean cells have above 0 concentrations even with the atmospheric corrections? What about the smearing error – couldn't that contribute by smearing out the ice into open water areas near the ice edge? The atmospheric correction is reducing the variability over open water but it does not totally remove it, and we are not correcting for cloud liquid water. Some of this residual bias can be regional leaving above 0% ice concentrations over open water in some places. This is not optimal, however, we wanted to avoid the use of weather filters with fixed thresholds. Weather filters with fixed thresholds are not consistent with our methodology (using dynamical tie-points and regional error reduction) and they may introduce artificial trends.

The smearing error is an issue near the ice edge and this is also evident in the uncertainty estimate. We have extended the discussion on these issues.

16/30: National Ice Center should be capitalized. Ok.

17/1: The missing chart period is noted for (at least) the third time. It's okay to reiterate the point for clarity, but each time it's written as if for the first time. At least rewrite to "As noted earlier, there were no digital ice charts available..." But I don't think it's needed at all here. Ok, it is deleted.

17/18: The overlap is during July and August, which isn't during the minimum. And I don't think the limited number of ice data points is the main factor here – it's that July and August is a time of significant melt and melt pond coverage. You are right. The sentence has been deleted and the possible geo-physical explanation given.

17/30: Repetitive of earlier description of the NIC charts – don't need to say it again here. Ok, it is deleted.

18/2: Who are the "two other producers"? Is this referring to other ice charting centers? Yes, it has been specified now (Norway and Greenland).

18/10-11: "...yield the same concentrations..." Ok.

18/30: Why is the 30% concentration used to define extent? It's perfectly reasonable,

but why use that instead of the more common 15% threshold? The reason for using 30% in this study is because it gave less noise in the example applications. When not using weather filters there is still a residual noise over open water.

19/16: The 5% and 95% ranges need to be explained. I would assume they describe the range between the highest and lowest 5% values. But then how in Figure 10 is the 2012 extent within the 5% contour when it is the lowest extent in the record? An explanation is included in the text. The variability that the 5 and 95 % are showing is the daily variability within each month of the record.

20/1: Recommend citing Parkinson, GRL, 2014, who calculated open water season trends for the Arctic from passive microwave sea ice fields. The reference has been included.

20/7-10: This sentence is a bit unclear. I think I understand – different periods were tested and 5 days was the shortest period found that gave reasonably stable values without much noise. Right? Right, this is now clarified in the text.

21/23-29: This seems odd to me. In Figure 14, the open water trend near the edge is +30 days for 2004-2014, but there have been record high extents during the same period? I guess I can see larger extents for shorter periods, but the large number seems unlikely – the open water season is over a month longer, but still ice reaches record maximums? Maybe this mostly points to the shortness of the period. Maybe it's just one year (or two) that are really anomalous that dominate the signal?

The significance levels on the short period indicates that the trend results are in general not significant and it is likely, as you say, that one or two unusual years dominate. However, for the long record most of the trends are significant (except perhaps along the ice edge) and the Bellinghausen Sea and East Antarctica is having more open water days and regions in the Ross and Weddell Seas are having less open water days.
The figures showing the short period (2004-2014) have been removed for the reasons that you mention.

Please also note the supplement to this comment:
http://www.the-cryosphere-discuss.net/tc-2016-34/tc-2016-34-AC2-supplement.pdf

---

## Author Comment (AC3) · 17 Jun 2016

Thanks for the constructive comments to our MS and helping us making this a better paper. We have tried to accommodate nearly all. A few of them would require significant redoing of the processing (e.g. the ice chart uncertainty estimation and the geo-location correction of all data) and so it is not possible now. However, we are working on new updates of the dataset and in next versions of the dataset we will try to implement your suggestions. See specific answers in the text below.
The manuscript presents a harmonized Arctic and Antarctic sea ice concentration data set based on passive microwave SMMR, SSM/I and SSMIS satellite observations from

[Figure]

1978 to 2014, spanning observations from 11 different instruments. The new things of this data set are the atmospheric correction and uncertainties given for each pixel individually, based on a careful error estimation. The various error components are considered separately and an overall error model is forwarded in Fig. a3. ËŸ

General comments: It is an elegant approach to use dynamic tie points to compensate for sensor drift and inter-sensor calibration. Were time series of the tie points also investigated? We have made detailed investigations of the tie-point window (7, 14 or 30 days) and we have also looked at the short overlap period between SMMR and SSMI. Looking at the tie-point time series has several difficulties because it is unclear if the variability that you see is due to geophysical changes, instrument drift or due to NWP model biases (if the atmospheric corrections has been applied). The link between the different noise sources and the sea ice concentration is indeed an active research topic.

Could a seasonal cycle, drift or jumps with sensor change be observed? What causes the tiepoint variability is difficult to assess (see above).

P(age)4L(ine)4 SSMIS data: While the SSM/I data from RSS comprise corrections for geo-location, C1 sensor calibration and inter-sensor calibration, this is not the case for the SSMIS data. Were similar corrections applied to the SSMIS data? No, we use the data as is.

Is the dynamic tie points method considered to compensate automatically these errors? The idea with the dynamical tie-points is to compensate the (unknown and) systematic errors in the retrieval process. Further, they make it possible to use NWP data for regional error reduction without introducing biases from the model. The random component of the tie-point is one of the two components in the sea ice concentration uncertainty estimate.

The large number of 14 subsections of Section 2 could be more structured for easy understanding. I would consider Secs. 2.6 to 2.10 as subsections of 2.5 Uncertainties.

According to the short Sec. 2.5 and Eq. (1) I would expect two subsections, one on the tie point uncertainty and one on the representativeness error, with perhaps more subsections. From the sentence in Sec 2.5 'The tie point uncertainty $\varepsilon$_tie-point , including residual atmospheric noise, sensor noise and ice surface emissivity variability, is derived from measurements as the first component of uncertainty' I would expect the section on the first component to treat exactly these errors in exactly this order. However, the heading of 2.6 is 'First component: instrument noise, algorithm and tie point uncertainties', which is different. If the error components are not treated in the mentioned order, then the relation between the different error contributions should be made clear. I do not find algorithm noise treated in Sec. 2.6 which is promised in the heading. Rather, is seems to be treated in Sec. 2.10. Moreover, Secs. 2.12 to 2.14 deal with the correction of known errors which could be a common heading for these three. We agree. We have followed your suggestions in the restructuring of section 2.

Sec. 1.5 Ice charts: It is very good to use ice chart as independent source. The data source should be indicated. Are those data freely available? Reference? Are they provided in the same polar stereographic grid as the SSM/I data? Is there any conversion required, e.g. from region shapes to pixel data? We have reprocessed the ESICR - ice chart comparison using publicly available and documented datasets. The data are from three data sources: 1) The northern hemisphere from 1972-2007 described in "National Ice Center. 2006. Updated 2009. National Ice Center Arctic sea ice charts and climatology. F. Fetterer and C. Fowler. Boulder. Colorado, USA. National Snow and Ice Data Center". 2) The Southern hemisphere from 1972-1994 described in F. Fetterer. A selection of documentation related to national ice center sea ice charts in digital format. NSIDC Special Report # 13, 2006. Boulder, Colorado, USA. 3) both the northern and southern hemisphere from 2006-2015 as shape files. The conversion to grid files is described in www.natice.noaa.gov/products/sigrid.html

These references and description of the data are included in the MS.

Some of the ice chart errors can be quantified: If the charts give ice concentrations

in 10% steps, this corresponds to adding quantization noise equally distributed in the IC interval [0,10]. It has a standard deviation of 10*sqrt(1/12) = 2.9% IC. Moreover, weekly ice charts will contain as error the development of the ice within one week. This can be estimated considering ice chart differences of successive weeks. This is a very good idea and we will try that in future comparisons. However, it has not been done here and the ice chart polygons do not have a standard interval of sea ice concentration, often it is given as 1-3 tens meaning that there are sections within the polygon where the sea ice concentration is between 10 and 30% (we interpret this as 20%). Some convention for translating the uncertainties needs to be developed. The sea ice services (Norwegian, Finnish and Greenlandic) are at the moment working on assessing the uncertainties in their ice charts. As far as I know there is no convention yet but I have forwarded your suggestion to them.

C2 Sec. 2.7 Geo-location error: A reference to Hollinger et al. (1990) is not suitable for SSMIS geolocation which was not launched at the time of publication. More suitable would e.g. be Poe et al., Geolocation Error Analysis of the Special Sensor Microwave Imager/Sounder, IEEE TGRS, VOL. 46, NO. 4, APRIL 2008. They find a geolocation error 'in excess of 20–30 km' which cannot be considered small compared to the footprint size. Was the SSMIS L2B NRT data version you are using (P4L4) ever corrected for such errors? Thanks for the reference it will be included together with Hollinger. Poe et al. describes a relatively simple method where the geolocation error can be reduced from "in excess of 20-30 km" in the cal/val phase to near 5 km. When the SSMIS L2B NRT dataset is used for sea ice drift analysis the standard deviation compared to drift-buoys is less than 4 km (http://osisaf.met.no/docs/osisaf_cdop2_ss2_valrep_sea-ice-drift-lr_v4p0.pdf). This is comparable to other sensors with comparable resolution (ASCAT, AMSR) and it shows that the SSMIS data geo-location accuracy is not in excess of 20-30 km it may rather be close to 5 km as indicated in the text. The 5 km is the value we used in the simulations to assess the impact of geo-location accuracy on the sea ice concentration estimate.

Anyway, the geolocation errors should be corrected beforehand. We are working together with the EUMETSAT climate SAF who are compiling different microwave radiometer data sets and geo-location correction is on top of our wish list. Geo-location is a prerequisite for doing proper land-spill-over correction.

It would be helpful for comparison with other publications (e.g. with Spreen et al. 2008: . J. Geophys. Res 113, C02S03, doi:10.1029/2005JC003384) to give, in addition to the time series of bias and stddev in Figs. 4-7, the average values for these quantities. We have recomputed the comparison to ice charts extending it to cover 1978-2015, but we did not find a way to include the absolute concentrations in the figure. In the ice charts the open water concentration is 0% and over ice the ice chart concentration is higher than the ESICR concentration.

It could be interesting to show in Figs. 8 and 9 also the open water bias for the overlap period. On average the open water bias is smaller for SMMR than for SSM/I because the SMMR 18.0 GHz channels is further away from the water vapour absorption line than the 19.3 GHz channel on the SSM/I. The overlap bias is shown geographically in fig. 15 and 16 in the validation report (http://osisaf.met.no/docs/osisaf_cdop2_ss2_valrep_ice-conc-reproc_v2p0.pdf) and will not be reproduced here.

Data Levels 3 and 4 at the beginning of Sec. 2.11 should be briefly explained. An explanation is included in the MS. Level 3 is left out because it is not needed here.

As TCD does no more any typesetting, it would be nice if the manuscript would obey simple rules of typesetting: numbered quantities are denoted with capital words like Figure 1, Table 3, etc., and numbers up to ten are written as words. The MS has been corrected with attention to typesetting.

Minor points: Use for the same quantity always the same symbol. Wind is sometime $u^*$, sometimes $U^*$. Ice concentration sometime ic, sometimes IC. Does the * symbol in Eq. (1) at $u^*$ have a meaning? We have been through the text carefully and corrected

this. u* is the friction velocity, which is the slope of the logarithmic wind profile (under certain conditions). The friction velocity is called u* in the literature and we will also use that term here.

P3L14 'There is SMMR data only every second day': No, each day. Only, because of the narrow swath, full coverage requires data accumulation over two days. Similar P15 L11. Actually the radiometer was only operated every second day to save power on the spacecraft working shifts with another instrument. This is now stated in the text.

P6L13 ' The representation of atmospheric liquid water column in the NWP data is not suitable to use for brightness temperature correction.': Give reason: because of the spatial and temporal variability of cloud, which is higher than the model grid cell size and model time step size. OK the reason has been included.

P7L1 '. The fluctuations due to atmospheric and surface emission are systematic.': Meaning unclear. It has been clarified.

P8 L7: explain OSI SAF It is spelled out in the abstract.

P9L23 'spatial ice concentration standard deviation': meaning of 'spatial' unclear. Omit? "spatial" has been omitted since it is clear from the rest of the sentence that it is the standard deviation of larger regions.

P11L14: meaning of 'logically' unclear. Omit? "logically" has been omitted.

P11L30 to P12L3: Necesscity of Eq.(6), Boxcar and Heaviside function unclear. For the subsequent text, it is sufficient to define the truncated ice concentration alpha by P12L4-6. This sections has been simplified omitting equations 6 and 7.

P13 L15 weighting function Eq. (12): In principle, the function should reflect the antenna pattern. Then it would have to be a Gaussian. Linear weights taken here as simple approximation. You are right and in coming versions of the dataset we will be using a Gaussian weighting function. However, for this dataset we use Eq. 12.

P14L22 and P15L2: equation 6 -> Eq. (13) OK

P15L18 Latter -> later OK, the validation has been extended and therefore the sentence omitted.

P19L22 '(Q3)' not needed, never used. OK, it has been deleted.

P20L1-5: There is an extended literature on detecting onset and end of melt season with sophisticated methods. It is ok to use a simple method here, but it should be mentioned that more exist, e.g. by citing the recent article by Close et al.: Regional dependence in the timing of onset of rapid decline in Arctic sea ice DOI: 10.1002/2015JC011187. See also literature cited there. The reference has been included for comparison.

P21L5: trend in number of open water days shown in figure 13 → Figure 12. OK.

P21 L13 . As for the Arctic the open water days is calculated.. → As for the Arctic, the open water days are calculated. OK.

P21 L 29 the pole → Antarctica or the South Pole We have changed it to Antarctica as suggested.

P23 L16 'The next update': insert 'of the ESICR data set' OK.

P23 L9 'minor differences': why should there be any differences between the two algorithms? Specify. The expected differences stem from the tie-point selection period which is either the last 30 days or 15 days before and after. This has been specified in the text.

Please also note the supplement to this comment:
http://www.the-cryosphere-discuss.net/tc-2016-34/tc-2016-34-AC3-supplement.pdf

---

## Author Comment (AC4) · 17 Jun 2016

[revised manuscript text omitted]

Formateret: Engelsk (USA)

For the Arctic there is a negative trend in the monthly mean extent for all months of the year (Table 3A). The negative slope is largest in September: -94 000±9700 km²/year and smallest in May: -32 000±4600 km²/year. The monthly trends for the Arctic are shown in Table 3A.

For the Antarctic there is a positive trend in the monthly mean extent for all months of the year (Table 3B). The positive slope is largest in April, October and December: at 33 000 km²/year and the smallest in February: 13 000±5400 km²/yr. The monthly trends for the Antarctica are shown in Table 3B.

Below we have looked at two periods of the 35 year ESICR: the entire 35-year period from autumn 1978 to the end of 2014 and the shorter recent 10 year period from 2004 to the end of 2014. The latter shorter period represents the period where most of the sea ice extent changes are taking place in both the southern and northern hemisphere.

Figure 10 shows tThe sea ice extent for the Arctic for both the long and the short records is shown in Figure 10 together with the September 2012 sea ice extent in Figure 10. The lower two panels are showingdisplay the seasonal variability of the sea ice extent and the long term mean monthly sea ice extent in March and in September. which is the months with maximum and minimum extent, respectively. In this panel we have included the extent for the most recent 11 year of ESICR (2004-2014) for comparison. September 2012 was the lowest sea ice extent on record in the Arctic since beginning of the satellite era. Over the 35 years of ESICR there is a negative trend in sea ice extent for all months of the year with the largest negative trend during the summer and the beginning of autumn

(Jul-Oct). i.e. the third quarter of the year (Q3).

[revised manuscript text omitted]

---

## Author Comment (AC5) · 17 Jun 2016

The comment was uploaded in the form of a supplement:
http://www.the-cryosphere-discuss.net/tc-2016-34/tc-2016-34-AC5-supplement.pdf

———————————————

---

## Author Comment (AC6) · 17 Jun 2016

The comment was uploaded in the form of a supplement:
http://www.the-cryosphere-discuss.net/tc-2016-34/tc-2016-34-AC6-supplement.pdf

———————————————

---

## Author Response (AR2)

Aug. 15., 2016

Dear Editor,

Thank you for your comments and sorry for not including answers to them in the first round. We have followed your recommendations and given a reply to each of the two comments below.

Best regards Rasmus

Editor Decision: Publish subject to minor revisions (Editor review) (08 Jul 2016) by Dr. Christian Haas

Comments to the Author:

Dear Authors,

thank you for the revisions. However, I think you haven't addressed my comments from the initial Editor's review. Please could you consider to include my suggestions, and to reply to them. My comments:

1. Given that sea ice coverage trends have been published previously and extensively, and are well established since many years, the motivation for your work is not really clear. I understand that your work is significant because it introduces new time series based on new algorithms, which are potentially superior to previously published work. I also understand that you may want to avoid statements as to which products are better, which is hard to convincingly prove anyways. However, I would like to request that you at least include comparison between the trends derived by you with previously published trends, and what the possible reasons for the agreements or disagreements are. Only then can the consequences and impacts of your work be evaluated, and why it would be worthwhile to reconcile studies of Arctic and Antarctic sea ice trends with your new product.

Reply:

Thanks for pointing this out. It is indeed important to reference ESICR to existing datasets but we will let the users of our dataset judge whether it is better than others. The motivation for this dataset is clearly described in the introduction, in particular points 1-4 describing a new methodology for processing sea ice concentration which is at the same time an answer to some problematic issues in the processing of previous datasets (1. sensitivity to atmospheric and surface emissivity trends: "artificial trends", 2. no noise reduction over both ice and water, 3. inter-sensor calibration issues, 4. no uncertainties). The fourth point: we have for the first time provided a dataset with spatially and temporally varying uncertainties along with the sea ice concentration. I think this is a convincing improvement compared to existing datasets and sufficient motivation for the reprocessing. In addition, as we mention in "future work" the dataset and the methodology for processing it are still being developed and this will be implemented in future updates.

Anyway, your comment is valid in order to relate ESICR to other datasets and we have therefore included a comparison between ESICR and the NSIDC sea ice extent and a discussion of the differences (new section 3.4 and an extra column in Tab. 3A and 3B).

2. A routine Similarity Assessment has revealed that your manuscript has a similarity index of 24% compared to previously published work. This is unusually high compared with other manuscripts. A closer

inspection revealed that approximately 20% of your manuscript was more or less copied literally from initial OSISAF reports (Tonboe and Nielsen, 2011; and Eastwood et al., 2010). As this is your own work, I think you cannot be accused of fraud; however, it would be important to cite those reports and to put your work into context and to point out the basis (and substantial historical experience) that has gone into it.

Reply:

Thanks, these two references appear together with the dataset at osisaf.met.no and of course these should also be included in the text and reference list of this MS. They are now included in the introduction explaining what they are and where to find them.

**The EUMETSAT sea ice concentration climate data record**

[revised manuscript text omitted]
 clockwisenorthward branches of the cyclonic atmospheric circulation along the ice edge.

In order to determineassess the periodlength of open watersthe ice season for a given pixel, the annual spatial distribution of dates of freeze-up and break-up were calculated using a simple methodology, yet the results are comparable to Parkinson (2014). The freeze-up date for a given point is defined as the date where the sea ice concentration climbs from below to aboveexceeds 30 % and remains so for at least 5 days. TheSimilarly, the break-up date for a given point is defined as the date where the sea ice concentration falls from above to below 30 % and remains so for at least 5 days. These thresholds were tested until a reasonable noise level was found.

The values for the ice concentration threshold and length of period were chosen by manually tuning for

convergence: ice concentrations lower than 30% and periods less than 5 days were found to produce

noise in the spatial distribution of freeze-up/break-up dates, which settles at the chosen values, though

somewhat less so in the short 10-year record.

Since the sea ice does not retreat and expand completely every year, not all areas experience the same number of freeze-ups and break-ups over an equal period of years. Therefore, some regions may experience relatively few freeze-ups and break-ups, thus reducing the confidence in the trend of the region. As a consequence, only areas having experienced more than 6 freeze-ups/break-ups in each period are considered.

Figure 12 is showing the decadal trend in open water days in the Arctic region covered by sea ice part

of the year. The open water days are calculated as the difference in days between freeze-up and breakup.

The  and the decadal trends in the open water days are shown in Figure 12 for both the longArctic and the short climate record in Figure 12 left and right, respectively14 for the Antarctic.

OverIn the longArctic, over the record of 35 years the ice season number of open water days has been shortenedincreasing 
[revised manuscript text omitted]

**Formateret:** Skrifttype: 10 pkt

**Formateret:** Indrykning: Venstre: 0 cm, Mellemrum Efter: 0 pkt.